# Immune-metabolic trajectories delineate subgroups in paediatric long COVID

Daniel Vilser [1,11], Irina Han [2,3], Katrin Vogel [2,3], Pauline Jakobs [2], Michael Lorenz [4], Peter Huppke [5,6], Lars Newman [1,5], Michelle Paszkier[2], Jens Kuhle[7], Juliane Mohr [3,8], Clara Aign[2], Annegret Reinhold [3,8], Dirk Reinhold [3,8], Stefan Weinzierl [9], Elisabeth Ullmann[1], Hans Proquitté [10] & Monika C. Brunner-Weinzierl [2,3] ✉

Most children and adolescents recover rapidly from SARS-CoV-2 infection, yet a subset develops paediatric long COVID (LC). How immune ontogeny shapes LC biology and heterogeneity remains unclear. We deeply phenotype a two-visit cohort with severe LC ($n = 74$) and controls ($n = 27$) spanning up to 3.2 years post index infection. Symptom burden remains high and neurofilament light chain (NfL) percentiles inversely associate with functional status (Bell score; $r = -0.3536$, $P = 0.0060$). Cardiopulmonary assessment and serology are unremarkable. Conventional autoantibodies are not enriched, whereas anti-DFS70 supports subgrouping. Immune features are temporally structured; SARS-CoV-2–associated mediators decline within 1 year, while innate-weighted, Th2-skewed cytokines persist. Metabolomics (43 metabolites) recapitulate the identified subgroups and align with EBV serostatus, disease phase (<1 year versus years 1–3.2), and anti-DFS70 positivity. In EBV-naïve LC, higher haemoglobin concentration (MCHC) tracks worse function, whereas higher IL-12p40, thiamine and basophils track milder impairment (all $P \le 0.0170$). These data delineate immune-metabolic and haematological axes of paediatric LC heterogeneity and support biomarker-guided stratification.

The clinical course of SARS-CoV-2 infection in children is typically benign and often asymptomatic, reflecting age-tuned immune regulation[1,2]. Nevertheless, approximately 1–3% of paediatric cases develop long COVID (LC; post-COVID condition), a debilitating syndrome persisting for ≥3 months without an alternative explanation[3–5], and appears less common than in adults[6]. Paediatric LC spans >200 symptoms across organ systems; ~20% of affected individuals report persistence beyond 1 year[5,7–10]. This breadth in presentation and trajectory argues for multiple endotypes rather than a single pathway. Risk factors include female sex, age >12 years, early variants,

[1]Post-COVID Outpatient Clinic, Department of Paediatrics and Adolescent Medicine, Jena University Hospital, Jena, Germany. [2]Department of Experimental Paediatrics, Otto-von-Guericke-University, Magdeburg, Germany. [3]ChaMP, Centre for Health and Medical Prevention, Otto-von-Guericke-University, Magdeburg, Germany. [4]Section of Paediatric Pulmonology, Department of Paediatrics and Adolescent Medicine, Jena University Hospital, Jena, Germany. [5]Department of Paediatric Neurology, Jena University Hospital, Jena, Germany. [6]Centre for Rare Diseases, Jena University Hospital, Jena, Germany. [7]Multiple Sclerosis Centre and Research Centre for Clinical Neuroimmunology and Neuroscience (RC2NB), Department of Neurology, Departments of Biomedicine and Clinical Research, University Hospital and University of Basel, Basel, Switzerland. [8]Institute of Clinical Immunology and Cell Therapeutics, Otto-von-Guericke-University, Magdeburg, Germany. [9]Audio Communication Group, Technische Universität, Berlin, Germany. [10]Section of Neonatology and Paediatric Intensive Care Medicine, Department of Paediatrics and Adolescent Medicine, Jena University Hospital, Jena, Germany. [11]Present address: Clinic of Paediatrics and Adolescent Medicine Ingolstadt/Neuburg, AMEOS Krankenhausgesellschaft Neuburg mbH, Neuburg, Germany. ✉e-mail: monika.brunner-weinzierl@med.ovgu.de

reinfection burden, and comorbidities. Despite growing recognition, the mechanisms sustaining paediatric LC and driving heterogeneity remain unclear; there are no pathognomonic findings or diagnostic tests, and management remains largely symptom-directed with limited mechanistic guidance[1,11,12].

Childhood and adolescent immunity are relatively enriched for tissue-repair and disease-tolerance programmes while maintaining efficient antiviral control, partly via locally enhanced IFN responses that limit immunopathology[13]. Consistent with this, outcomes of SARS-CoV-2 infection in this age group range from generally benign acute courses to hyperinflammatory entities such as MIS-C and related post-infectious syndromes, underscoring a distinct clinical–immunological entity[3,14]. Across ages 7–19 years, puberty-associated thymic decline and hormonal shifts narrow, but do not erase, paediatric advantages relative to adults, potentially reshaping the balance of immune axes and tissue-resident programmes underlying endotypic heterogeneity[15–18]. With fewer accumulated immunological 'imprints' (for example EBV latency), paediatric LC provides a tractable setting to dissect core protective and pathogenic programmes underlying fatigue, cognitive and dysautonomic symptom clusters, insights that may generalise across ages[19–23]. Autoimmune phenomena have been reported in paediatric LC, but population-level signals for overt autoimmune disease are modest and not uniformly specific[10,24]. Together, these findings suggest that subsets of paediatric LC show immune dysregulation and autoreactivity signatures without consistent overt systemic autoimmunity. Current evidence implicates altered immune responses, cellular metabolism and endothelial function[3], yet whether these processes segregate into coherent endotypes and how they evolve over time remains unresolved.

We therefore hypothesised that paediatric LC comprises temporally dynamic biological subgroups. To address this, we performed biologically motivated analyses spanning disease phase, immune imprinting, autoreactivity, immune polarisation, and organ-specific involvement, including the possibility that waning SARS-CoV-2-associated antiviral and humoral features contribute to subgroup structure over time. Specifically, we aimed to identify temporally structured immune subgroups across the paediatric LC course, relate these to recovery trajectories, and test whether biologically motivated stratifiers, including markers of immune imprinting and autoreactivity, distinguish distinct biologic signatures and clinical severity. A subgroup-specific understanding, with a particular focus on protective mechanisms, could help guide future research, enabling precision interventions and improving the design and outcomes of clinical trials in both paediatric and adult LC[25].

## Results

To interrogate paediatric LC heterogeneity, we analysed biologically motivated subgroups based on disease phase, EBV exposure, immune polarisation, and organ involvement, and assessed complementary clinical, immunological, metabolic, and functional readouts within each subgroup. We analysed 74 children with active LC symptoms and 27 controls, defining the index date as the last documented SARS-CoV-2 infection preceding LC onset; time since index infection at study visit (TsinceIndex) ranged up to 166 weeks (-3.2 years) (Fig. 1a–e; Supplementary Table 1a, b). Controls comprised healthy children ($n = 14$) and clinically stable children with cystic fibrosis (CF; $n = 13$), included as a respiratory infection-exposed comparator group[26]. As pre-specified, both control subgroups were analysed as a single control arm; baseline inflammatory profiles were comparable between healthy and CF controls (Supplementary Table 1c), supporting pooling for subsequent analyses. Controls were slightly younger (median difference of 3.1 years). LC participants were re-assessed 3–6 months later (LC visit 2, follow-up). Symptom burden and functional impairment were quantified at each visit using validated patient- and parent-reported instruments and performance tests (Fig. 1e; Supplementary Fig. 1)[27].

As an a priori marker of antiviral activity, we measured serum IFNα. IFNα was elevated in paediatric LC within the first year after the index infection compared with controls, whereas complement parameters, including C3, C4, and total haemolytic complement activity (CH50), remained within age-adjusted laboratory reference ranges across the observation period (Fig. 1d). Beyond 1 year of LC, IFNα levels were significantly lower (estimate = −6.02, 95% CI: −10.96 to −1.09; $F(1, 95.39) = 5.83$, $P = 0.0177$; $R^2m = 0.0196$), consistent with waning antiviral signalling at later disease stages.

Patients assessed within 1 year of active LC showed no group-level improvement in physical or mental health measures (Fig. 1c, e; linear mixed-effects models (LMMs) details in Supplementary Fig. 1a, b and Supplementary Table 1b). Across the broader post-infection window (up to 3 years), we similarly observed no consistent improvement at the cohort level, although individual trajectories varied, with some patients showing modest improvement or decline. We next tested whether previously proposed risk factors for paediatric LC were associated with symptom severity. We fitted a pre-specified LMM with patient ID as a random intercept and included comorbidity status, sex and SARS-CoV-2 variant-of-concern (VOC) wave as fixed effects, adjusting for age, number of vaccinations and number of infections prior to LC onset (Supplementary Table 1d, lower panel). Across multiple pre-defined severity readouts, including the sit-to-stand test and questionnaire-based measures such as the Bell score, we found no evidence that these variables were associated with disease severity in this cohort (all $P > 0.6$). Thus, within our study, these proposed risk factors did not explain inter-individual differences in the severity of the LC phenotype.

Given the high prevalence of fatigue and dyspnoea in LC, we performed a protocolised assessment of potential cardiac and pulmonary involvement in paediatric LC (Figs. 1c and 2a, b)[8,28]. All LC participants ($n = 74$) underwent a 12-lead electrocardiogram (ECG) and transthoracic echocardiography at presentation. ECG showed supraventricular extrasystoles in one participant. Echocardiography revealed findings in three patients (4.1%): mild mitral regurgitation; a bicuspid aortic valve with a coronary artery fistula; and mild diastolic dysfunction that resolved at follow-up. Paediatric cardiology review deemed these findings incidental and not explanatory of the LC phenotype.

Pulmonary involvement was assessed using FEV1 $z$-scores standardised to paediatric reference values. Values were largely within the normal range, with only a minority at or below the lower limit of normal ($z \le -1.64$) (Fig. 2b), arguing against a cohort-level obstructive ventilatory defect. In an LMM accounting for repeated measures (random intercept: patient ID), FEV1 $z$-scores were not associated with comorbidity status, age, sex, VOC wave, number of infections prior LC onset, vaccination prior LC onset, or TsinceIndex window (<1 year vs 1–3 years) (Supplementary Table 2.1).

We next tested an a priori immunological framework centred on epithelial/type-2 signalling implicated in airway biology (IL-33, IL-13)[29] and systemic inflammation (IL-6) (Fig. 2c). Within the first year after TsinceIndex (index infection preceding LC onset), IL-13 and IL-33 were significantly higher in paediatric LC compared with controls after Holm–Bonferroni adjustment ($P < 0.05$), whereas IL-6 was not systemically elevated. We then modelled $FEV_1$ $z$-scores using LMM, including TsinceIndex and the selected cytokines (Fig. 2d; Supplementary Table 2.2). To avoid multicollinearity and maintain model interpretability, we applied a pre-specified rule to retain one representative of highly correlated predictors ($r > 0.7$); accordingly, IL-13 was kept as the representative type-2 effector signal, and IL-33 was excluded from the final model (Supplementary Table 2.2)[15,16]. The final model explained 16% of the variance through fixed effects ($R^2m = 0.1622$), whereas $R^2c$ (0.7210) highlighted substantial between-individual variability captured by the random intercept. Additional adjustment for the pre-specified covariates did not materially change

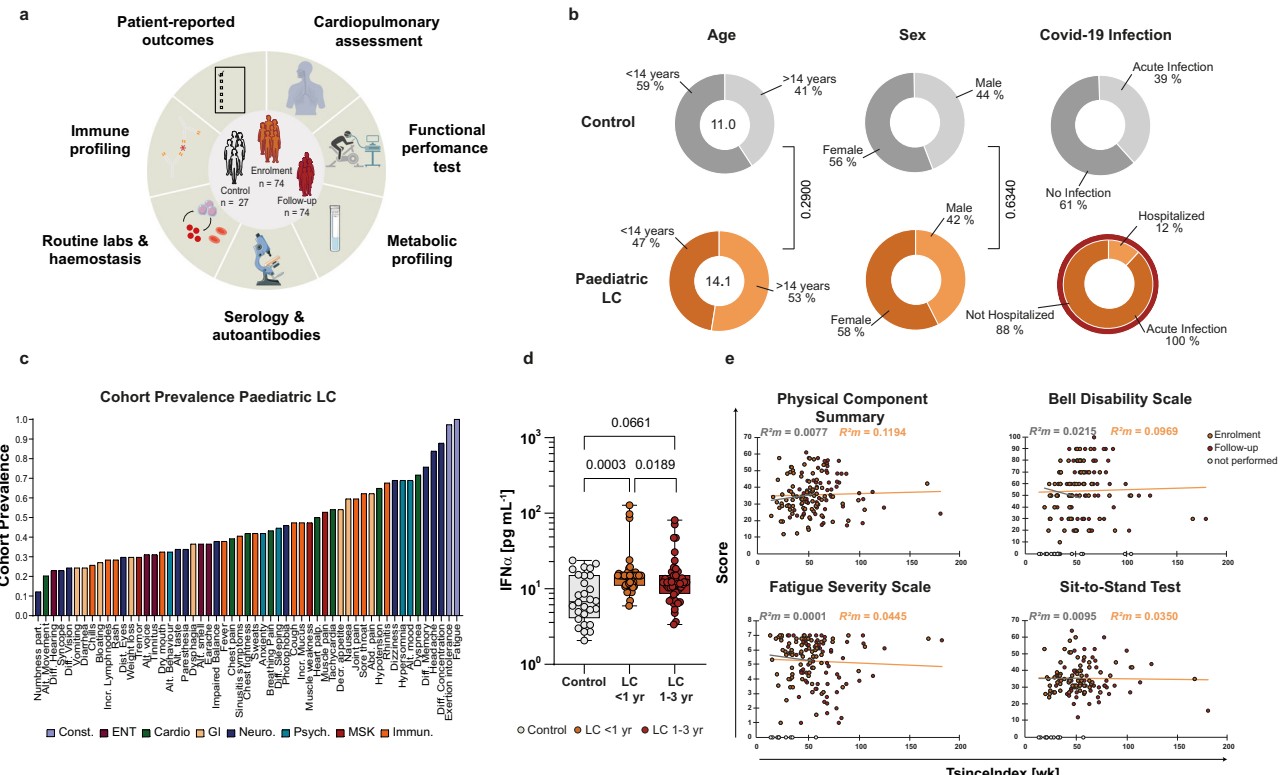

**Fig. 1 | Demographic and clinical characterisation of paediatric participants.**
**a** Study design and phenotyping overview. Controls (*n* = 27, grey) had no clinical diagnosis of long COVID (LC). LC participants were assessed at enrolment (*n* = 74, orange) and re-assessed 3–6 months later (*n* = 74). Created in BioRender. Ja, P. (2026) https://BioRender.com/5hr24rq. **b** Group demographics (age, sex, and COVID-19 infection characteristics). Age category (0–14 vs. >14 years) and sex were compared between groups using two-sided Pearson's $\chi^2$ tests, with Holm–Bonferroni-adjusted *P* values reported. Neither age category ($\chi^2(1) = 1.13$, *p* = 0.2877, Cramér's *V* = 0.1000) nor sex distribution ($\chi^2(1) = 0.23$, *p* = 0.6316, Cramér's *V* = 0.0474) differed significantly between groups. **c** Symptom prevalence at enrolment, ranked by frequency and grouped by system. **d** Serum IFNα in controls (*n* = 27, grey) and paediatric LC stratified by TsinceIndex. Box plots show median (IQR); whiskers indicate min–max. For between-group comparisons, repeated measurements were averaged within each time window to one value per individual (<1 year, *n* = 47, light orange; 1–3 years, *n* = 45, dark orange). Group differences were

assessed using a two-sided Kruskal–Wallis test, followed by adjusted Dunn's post hoc pairwise comparisons versus controls (*P* = 0.0002; *P* = 0.0220, respectively). Within-LC time-window comparisons were assessed using a two-sided linear mixed-effects model (LMM; *n* = 139 observations; *P* = 0.0189) with TsinceIndex window as a fixed effect (*P* = 0.0177) and participant ID as a random intercept. **e** Physical Component Summary (PCS; *n* = 143 observations), Bell disability scale (*n* = 121 observations), Fatigue Severity Scale (FSS; *n* = 139 observations) and sit-to-stand score (*n* = 120 observations) versus TsinceIndex were assessed using two-sided LMMs with participant ID as a random intercept; points denote individual observations; lines are shown for visual guidance only. Black and red lines indicate descriptive regression lines for the respective intervals. Outcome-specific model outputs are provided in Supplementary Fig. 1 and Supplementary Table 1b. For the four outcomes shown here, Holm–Bonferroni-adjusted *P* values were 1.0. For (**b**, **d**, **e**), two-sided *P* values were adjusted for multiple comparisons using the Holm–Bonferroni method.

the associations (Supplementary Table 2.2b). Notably, higher FEV1 *z*-scores were associated with lower IL-6 and higher IL-13, consistent with concurrent pro-inflammatory signalling and type-2–linked tissue-remodelling programmes potentially contributing to inter-individual heterogeneity in pulmonary function.

To characterise systemic immune activation in paediatric LC during the pre-specified primary analysis window of the first year after the LC-index SARS-CoV-2 infection, we quantified circulating serum cytokines and grouped them into pre-defined functional families, each represented by an a priori selected lead cytokine (SARS-CoV-2-associated, Th1/Th2 balance, innate-like, Th17/Th22-based and regulatory-like; Table 1, Supplementary Fig. 2). These cytokine family analyses within the first-year window constituted the pre-specified primary cytokine objective. Multiple testing was controlled within each family (Supplementary Table 2.3). The 1-year window was chosen to maximise comparability across studies and to capture the phase in which SARS-CoV-2-specific immune signals are expected to wane[28,30].

Within the SARS-CoV-2-associated family, the lead cytokine IL-13 was elevated in LC within the first year and declined thereafter to levels comparable to controls (Fig. 2c). IL-33 showed a similar transient

increase, whereas IL-6 did not differ significantly from controls. Based on prior reports in young adults[31], we additionally assessed anti-CCP and anti-Transglutaminase (anti-TransG) autoantibodies; both remained within the reference range and were not increased in paediatric LC compared with pooled controls (Fig. 2e; Supplementary Table 3.3a). These findings were robust in sensitivity analyses evaluating healthy and CF control subgroups separately against LC (Supplementary Table 3.3b, c; Supplementary Fig. 4).

Across families, additional pre-specified lead markers were significant within the first year, including the IL-4/IFNγ ratio (Th2/Th1 balance), IL-1β (innate-like), IL-11 (regulatory-like) and IL-12p40 (Th17/Th22-based) (Fig. 2f; Table 1). Family-level analyses revealed increased Th2-associated cytokines (IL-4 and IL-5), whereas Th1-associated cytokines (IFNγ, IL-2 and IL-12p70) were not elevated (Supplementary Fig. 2a, b). Among innate-like cytokines, IL-1α increased while GM-CSF was unchanged (Fig. 2g; Supplementary Fig. 2c). Within the regulatory-like family, IL-11 increased, whereas IL-10, IL-15, IL-18 and IL-27 did not (Fig. 2h; Supplementary Fig. 2c). Within the Th17/Th22-based family, IL-12p40 remained significant after Holm–Bonferroni adjustment (adjusted *P* = 0.048), while IL-17A, IL-17F, IL-22 and IL-23 did not (Fig. 2i; Supplementary Table 2.3e). Collectively, these data

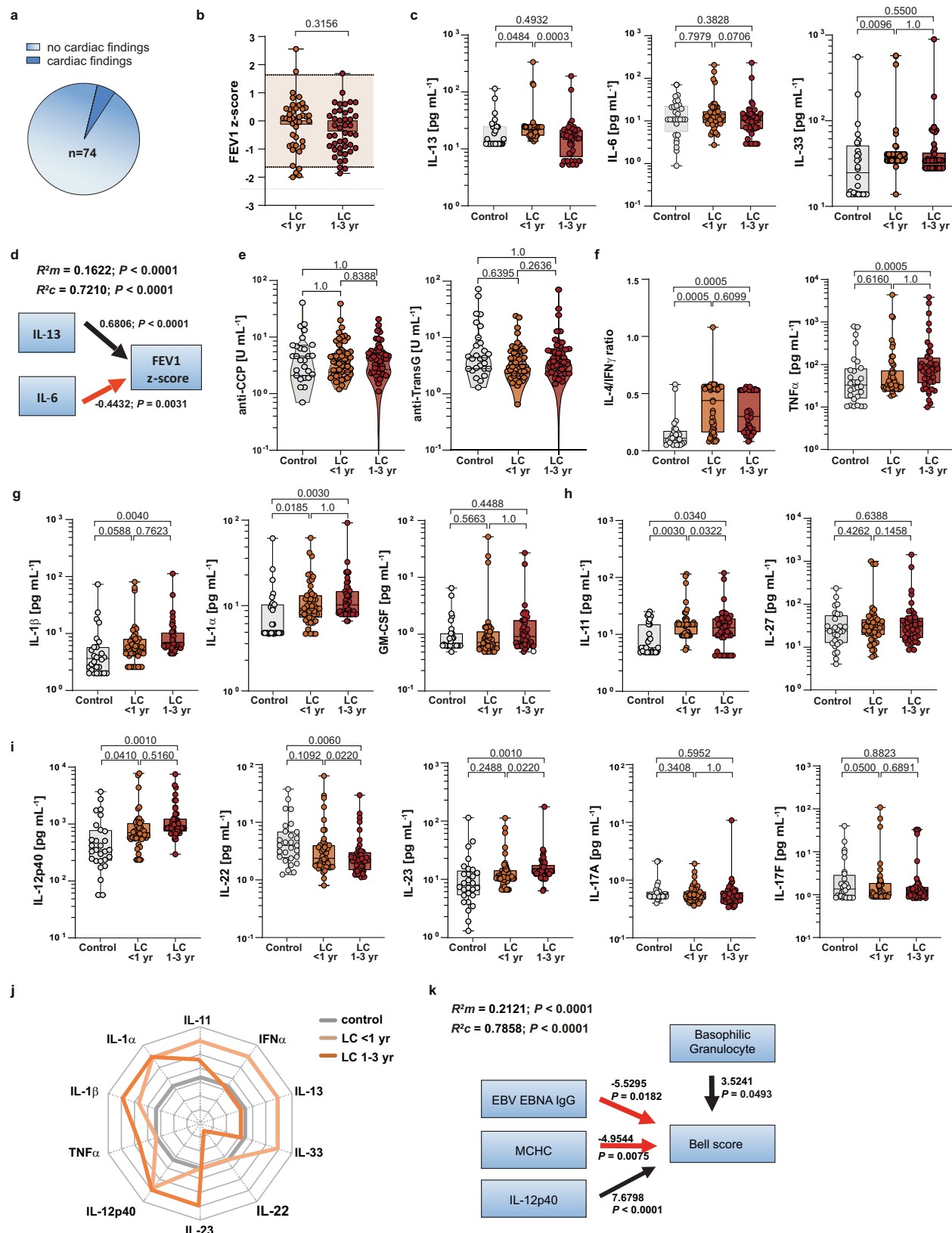

define a first-year LC signature characterised by immunological dysregulation with distinct cytokine patterns.

## Paediatric LC beyond the first year

Subsequently, we monitored systemic parameters after 1 year of LC to delineate the evolving immunopathological phases. Therefore, we

performed two complementary analyses: a cross-sectional comparison of cytokine levels in the LC 1–3 year group versus controls (Fig. 2c, f–i; Supplementary Fig. 2; Supplementary Table 2.3 for within-family multiple comparisons), and a LMM within the LC cohort with time window based on TsinceIndex (<1 year vs 1–3 years) included as a fixed effect and participant ID included as a random intercept to account for

**Fig. 2 | Immune, pulmonary and endothelial features of paediatric LC. a** Pie chart illustrating the prevalence of cardiac findings in paediatric LC ($n = 74$ biologically independent participants), including abnormal electrocardiography and/or echocardiography. **b** Box plots show forced expiratory volume in 1 s (FEV1) $z$-scores as participant-level summaries per TsinceIndex window (<1 year, $n = 42$, light orange; 1–3 years, $n = 44$, brown). The grey band denotes the normal reference range (±1.64). Differences across TsinceIndex windows were assessed using a two-sided LMM ($n = 133$ observations; $P = 0.3156$) with TsinceIndex window as a fixed effect and participant ID as a random intercept. **c** Systemic cytokine levels (IL-13, IL-6 and IL-33) in controls ($n = 27$, grey) and paediatric LC stratified by TsinceIndex; repeated measurements were averaged within each TsinceIndex window to one value per individual (<1 year, $n = 42$, light orange; 1–3 years, $n = 44$, dark orange). Group differences were assessed using a two-sided Kruskal–Wallis test, followed by unadjusted Dunn's post hoc pairwise comparisons versus controls; $P$ values were adjusted using Holm–Bonferroni. Within-LC time-window comparisons were assessed using a two-sided LMM ($n = 139$ observations) with TsinceIndex window as a fixed effect and participant ID as a random intercept. **d** Multivariable two-sided LMM for FEV1 $z$-scores ($n = 133$ observations) showing associations with IL-13 and IL-6, with participant ID as a random intercept; model fit is reported as marginal and conditional $R^2$ (see Supplementary Table 2.2). **e** Autoantibody readouts (anti-CCP

and anti-TransG) in controls ($n = 27$, grey) and paediatric LC stratified by TsinceIndex; repeated measurements were averaged within each TsinceIndex window to one value per individual (<1 year, $n = 47$, light orange; 1–3 years, $n = 46$, dark orange). Group differences were assessed using a two-sided Kruskal–Wallis test, followed by adjusted Dunn's post hoc pairwise comparisons versus controls. Within-LC time-window comparisons were assessed using two-sided LMM ($n = 139$ observations) with TsinceIndex window as a fixed effect and participant ID as a random intercept. **f–i** Additional systemic cytokines across controls (grey) and paediatric LC (<1 year, light orange; 1–3 years, dark orange) stratified by TsinceIndex; statistical analysis as in (**c**). The full cytokine panel and multiple-comparison results (Holm–Bonferroni) are provided in Supplementary Fig. 2 and Supplementary Tables 2.3 and 2.4, and for full autoantibody results in Supplementary Table 3.3. In (**b**, **c**, **e–i**), box plots show median (IQR); whiskers indicate min–max. **j** Radar plot summarising cytokines that differed significantly across LC TsinceIndex windows relative to controls (ordinal coding: no significant difference; significant increase; further increase versus LC < 1 year). **k** Multivariable Bell score model (two-sided LMM; participant ID as a random intercept). Full model output, including $P$ values, is provided for (**d**) in Supplementary Table 2.2 and for (**k**) in Supplementary Table 2.5.

## Table 1 | Lead cytokines across pre-defined functional cytokine families in paediatric LC

| Functional families | Lead cytokines | Control vs. LC < 1 yr | | | | Control vs. LC 1–3 yr | | | |
|---|---|---|---|---|---|---|---|---|---|
| | | **P** | **Z** | **ES** | **Adj. P** | **P** | **Z** | **ES** | **Adj. P** |
| SARS-CoV-2-related | IL-13 | 0.0242 ↑ | 2.508 | 0.2894 | **0.048** | 0.2466 ↔ | 1.159 | 0.4926 | 0.2466 |
| Th2/Th1 balance | IL-4/IFNγ ratio | <0.0001 ↑ | 5.222 | 0.6070 | **0.0005** | <0.0001 ↑ | 4.070 | 0.4796 | **0.0005** |
| Innate-related | IL-1β | 0.0294 ↑ | 2.178 | 0.2515 | **0.0294** | 0.0020 ↑ | 4.757 | 0.5570 | **0.0340** |
| Regulatory-related | IL-11 | 0.0010 ↑ | 3.500 | 0.4041 | **0.0040** | 0.0170 ↑ | 2.386 | 0.2793 | **0.0340** |
| Th17/22-related | IL-12p40 | 0.0102 ↑ | 2.570 | 0.2969 | **0.0306** | 0.0002 ↑ | 4.827 | 0.5650 | **0.0008** |

Pre-defined lead cytokines representing each cytokine family (SARS-CoV-2-associated, Th2/Th1 balance, innate-like, regulatory-like, and Th17/Th22-based) are shown for comparisons between controls ($n = 27$) and paediatric LC stratified by TsinceIndex (<1 year, $n = 47$; 1–3 years, $n = 45$). Group differences were assessed using a two-sided Kruskal–Wallis test, followed by an unadjusted Dunn's post hoc test versus controls. $P$ values from Dunn's tests are two-sided. Holm–Bonferroni adjustment for multiple comparisons was applied across the five lead cytokines, and adjusted $P$ values are shown. Arrows indicate the direction of change relative to controls (↑ higher, ↓ lower, ↔ no difference).
LC long COVID, yr year, TsinceIndex time since index SARS-CoV-2 infection, P P value, Z Dunn's Z statistic, ES effect size, Adj. P adjusted P value (Holm–Bonferroni adjusted).

repeated measures (Fig. 2c, f–i; Supplementary Table 2.4). Lead-cytokine comparisons are summarised in Table 1. The previously elevated IFNα levels declined, indicating a waning antiviral response (Fig. 1d), in parallel with a decrease in the lead cytokine IL-13 within the SARS-CoV-2-associated cytokine family (Fig. 2c). In addition, the innate-like lead cytokine IL-1β was significantly upregulated, as was IL-1α within this family, but not GM-CSF. Within the Th17 axis, lead IL-12p40 and IL-23 were increased, but IL-22 significantly decreased after 1 year of LC. Still, the IL-4/IFNγ ratio (Fig. 2f) was increased and remained elevated throughout, suggesting a transition towards a Th2/17-driven state. Taken together, these findings outline an early antiviral and Th2-skewed response within the first year shifts towards a waning SARS-CoV-2 response showing innate- and Th2/17-oriented responses. This is consistent with the progressive immunopathological remodelling associated with chronicity (Fig. 2j).

Next, we fitted a pre-specified LMM (random intercept: patient ID) to quantify determinants of LC severity, using the Bell score as the dependent variable (Fig. 2k; Supplementary Table 2.5). We hypothesised that functional impairment in paediatric LC reflects sustained immune activation and granulocyte perturbations, and that prior EBV infection may modify these associations through immune imprinting. Accordingly, covariates were defined a priori to represent complementary biological axes rather than being selected by univariable screening: a marker of prior EBV exposure (anti-EBV EBNA; exposure history, not reactivation), mean corpuscular haemoglobin concentration (MCHC)[32], as a haematological readout, IL-12p40 as a robust proxy of IL-12/IL-23–axis activity, and basophil granulocyte counts as an indicator of granulocyte lineage shifts. The fixed effects explained 21%

of Bell score variance ($R^2m = 0.2121$), and each covariate was significantly associated with the Bell score. Notably, anti-EBV EBNA and MCHC decreased with clinical improvement, whereas basophil counts and IL-12p40 increased. To assess robustness and potential effect modification, we next fitted an expanded model that adjusted for key confounders (age, sex, comorbidity, and TsinceIndex; Supplementary Table 2.5b). Based on accumulating evidence implicating mitochondrial dysfunction in LC and the essential role of thiamine (vitamin B1) in mitochondrial energy metabolism[33,34], we additionally included vitamin B1 as a pre-specified metabolic covariate. Finally, to directly test the hypothesis that prior EBV exposure modifies specific biological correlates of severity, we incorporated a priori anti-EBV EBNA interaction terms. In this fully adjusted interaction model, most interaction terms were not statistically significant; however, anti-EBV EBNA showed significant interactions with MCHC and vitamin B1 (Supplementary Table 2.5b), motivating targeted follow-up analyses stratified by EBV exposure.

**Autoantibodies (aAb) and organ Injury in children with active LC**
To further test the hypothesis that paediatric LC comprises biologically heterogeneous strata, including subsets with potential CNS-related symptomatology, we measured serum neurofilament light chain (NfL) as a pre-specified exploratory circulating marker (Fig. 3a; Supplementary Fig. 3a). In parallel, we assessed disease-associated autoantibody profiles as a pre-specified autoantibody axis to evaluate autoreactive immune signatures. NfL was interpreted as a hypothesis-generating marker of potential neuro-axonal injury. NfL concentrations were converted to age-adjusted $z$-scores using the Basel

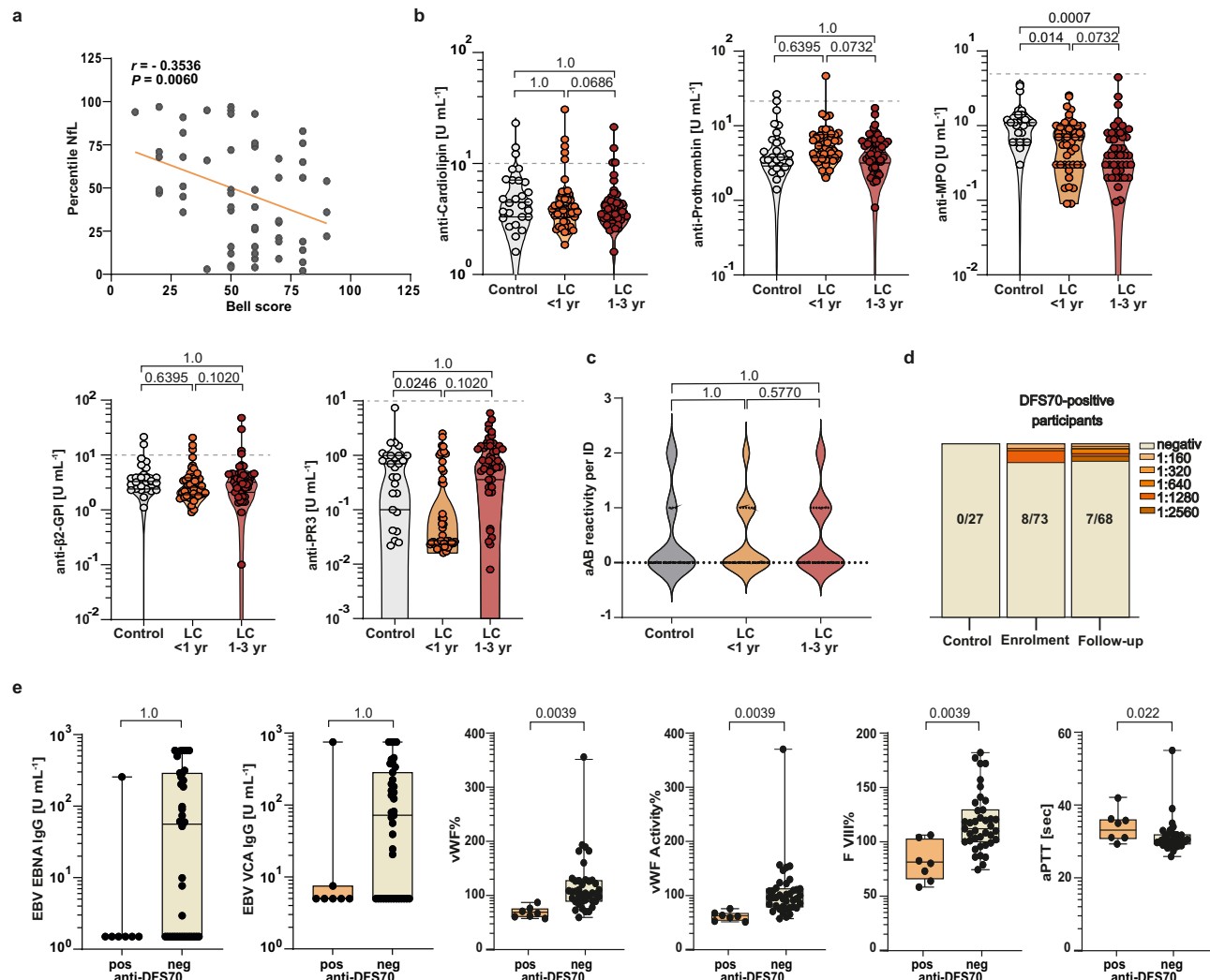

**Fig. 3 | Systemic autoantibody profiles and endothelial features of paediatric LC. a** Serum NfL percentile versus Bell score in LC at enrolment; each dot represents one participant ($n = 59$). Pearson's $r$, two-sided $P$ value and the fitted regression line are shown. **b** Autoantibody (aAb) readouts in controls ($n = 27$, grey) and paediatric LC stratified by TsinceIndex; repeated measurements were averaged within each TsinceIndex window to one value per individual (<1 year, $n = 47$, light orange; 1–3 year, $n = 46$, dark orange)(Supplementary Table 3.3). Group differences were assessed using a two-sided Kruskal–Wallis test, followed by adjusted Dunn's post hoc pairwise comparisons versus controls. Within-LC comparisons over the complete time course were assessed using two-sided LMM ($n = 139$ observations) with TsinceIndex window as a fixed effect and participant ID as a random intercept (Supplementary Table 2.4f). **c** Total aAb reactivity in controls ($n = 27$, grey) and LC stratified by TsinceIndex; repeated measurements were averaged within each TsinceIndex window to one value per individual (<1 year, $n = 47$, light orange; 1–3 years, $n = 46$, dark orange). Group differences were assessed using a two-sided

Kruskal–Wallis test, followed by adjusted Dunn's post hoc pairwise comparisons versus controls (Supplementary Table 2.3). Within-LC time-window comparisons were assessed using a two-sided LMM ($n = 139$ observations) with TsinceIndex window as a fixed effect and participant ID as a random intercept (Supplementary Table 2.4). **d** Proportion of anti-DFS70-positive participants in controls at enrolment and at follow-up (0/27, 8/73 and 7/68, respectively). **e** Coagulation parameters and anti-EBV antibody levels in anti-DFS70-positive versus anti-DFS70-negative LC participants. For visualisation, repeated measurements within the LC < 1-year TsinceIndex window were averaged to one value per individual (anti-DFS70-positive, $n = 7$, light orange; anti-DFS70-negative, $n = 39$, beige). For statistics, all available observations ($n = 139$) were analysed using two-sided LMMs; additional parameters, exact $P$ values and multiple comparison using Holm–Bonferroni are shown in Supplementary Table 3.4 and Supplementary Fig. 3c–e. In (**b**, **c**, **e**), box plots and violin plots show median (IQR); whiskers indicate min–max.

paediatric reference dataset and expressed as percentiles relative to healthy children[35]. In a one-sample comparison against the reference median (P50), we did not observe a significant upward shift at the cohort level ($W = 1084$, $n = 73$, $P = 0.20$; $r = 0.15$). A subset of children fell into the upper tail of the age-referenced distribution (12.3%; 9/73 > P90; Supplementary Fig. 3a), consistent with inter-individual heterogeneity rather than a uniform cohort-wide shift. In exploratory analyses, NfL percentile was inversely associated with functional status (Pearson's $r = -0.3536$, $P = 0.0060$; Fig. 3a). Using a clinically anchored threshold for severe impairment (Bell score ≤40), children below this threshold displayed higher NfL percentiles ($P = 0.0058$;

Supplementary Fig. 3a, left). NfL percentiles were not associated with TsinceIndex ($P = 0.8727$; Supplementary Fig. 3a, right). Collectively, these findings indicate heterogeneity in NfL and an exploratory association with clinical impairment, motivating mechanistic follow-up to clarify drivers and clinical relevance.

In separate analyses, to probe a potential autoimmune contribution to autonomic involvement, aAbs against GPCR (GPCR-aAb) targeting β1/β2-adrenergic and M3/M4-muscarinic receptors were quantified by ELISA (Supplementary Fig. 3b). LMMs showed that these aAb were not associated with the pre-specified early TsinceIndex window (≤1 year), systemic cytokine levels, or disease severity

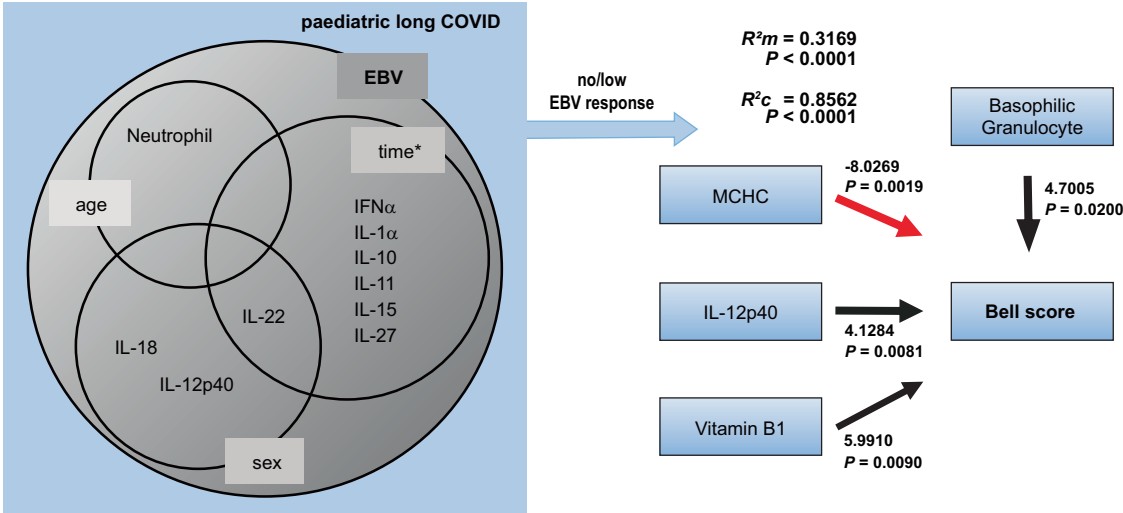

* TsinceIndex

**Fig. 4 | Biological EBV-linked subgroups of paediatric LC.** *Left*, Graphical summary of two-sided LMMs (Table 2; Supplementary Tables 4 and 5). Only associations that remained significant after Holm–Bonferroni adjustment across 32 dependent variables are shown; the outer grey circle denotes outcomes significantly associated with EBV exposure. Overlaps indicate outcomes additionally associated with the indicated covariates (TsinceIndex, age, sex, vaccination, and comorbidity). Repeated measures were accounted for by including participant ID as a random intercept. *Right*, Within the EBV-naïve/low subgroup (anti-EBV EBNA IgG <50 U mL$^{-1}$), associations of Bell score with basophilic granulocytes, MCHC, IL-12p40 and vitamin B1 were assessed using a two-sided LMM (participant ID as a random intercept; Supplementary Table 6). The model additionally included age, sex, TsinceIndex, vaccination and comorbidity as covariates.

(Supplementary Table 3.1). Moreover, aAb levels did not differ between LC TsinceIndex windows (<1 year vs 1–3 years; Supplementary Fig. 3b; Supplementary Table 3.2). Next, we analysed additional aAbs as surrogate markers of vasculitis (anti-proteinase 3 (PR3) and anti-myeloperoxidase (MPO)) and antiphospholipid syndrome (anti-cardiolipin and anti-β2-glycoprotein I (anti-β2GPI)) (Figs. 2e and 3b; Supplementary Table 3.3). None of these markers, including anti-CCP and anti-Transglutaminase (anti-TransG) assessed in Fig. 2e, were elevated relative to controls, and all remained below the assay cut-off during the first year of paediatric LC (Supplementary Table 3.3a). These null findings were unchanged in sensitivity analyses using healthy and clinically stable cystic fibrosis controls as separate comparator groups (Supplementary Table 3.3b, c; Supplementary Fig. 4). Within-LC comparisons using LMMs, with TsinceIndex window as a fixed effect and participant ID as a random intercept, likewise showed no increase in aAb levels between LC <1 year and LC 1–3 years (Figs. 2e and 3b and Supplementary Table 2.4f). To exploratorily assess whether enhanced autoreactivity is generally present in active paediatric LC, we quantified the prevalence of aAb positivity across individuals (Fig. 3c; Supplementary Tables 2.3 and 3.3) and observed no significant increase compared with controls, arguing against an aAb-mediated organ-injury phenotype in this cohort.

Within the pre-specified aAb axis, we additionally evaluated isolated anti-DFS70 reactivity (anti-DFS70 (LEDGF/p75) aAbs) as a potential marker of a distinct, potentially benign autoreactivity signature (Fig. 3d). Notably, none of the controls were anti-DFS70-positive, whereas 11% of patients with active paediatric LC tested positive; anti-DFS70 positivity persisted at a follow-up visit months later. We therefore stratified LC patients by anti-DFS70 status (Fig. 3e; Supplementary Fig. 3c–e; Supplementary Table 3.4). Within the LC cohort, anti-DFS70-negative patients showed higher von Willebrand factor activity/frequency and elevated factor VIII levels, while other haemostatic parameters (antithrombin III, fibrinogen, D-dimer, aPTT, protein C, free protein S) and complement components (C3, C4) remained unchanged. Because the extended coagulation panel was not available for controls, these analyses reflect within-cohort stratification rather

than case–control differences. EBV-related antibodies were independent of anti-DFS70 status in LC patients. Together, these data indicate a coagulation-factor signature in a defined LC subgroup and suggest that anti-DFS70 reactivity, classically associated with non-systemic autoimmunity, may capture a distinct, potentially less-pathogenic immune state in paediatric LC[36].

## Molecular landscape of EBV-associated paediatric LC

EBV has been implicated in post-viral immune dysregulation and is a plausible modifier of paediatric LC biology. We therefore assessed whether EBV exposure contributes to the molecular and clinical landscape of paediatric LC by modelling EBV serostatus (anti-EBV EBNA binary) as a covariate in LMMs, with repeated measures clustered by patient ID, across pre-defined mechanistic readout families, adjusting for sex, age, TsinceIndex, vaccination status and comorbidity (Fig. 4, left; Table 2; Supplementary Table 4). Multiple testing was controlled within each family (Holm–Bonferroni), and the composite summary highlights only readouts with significant overall model fit and a significant positive contribution of EBV exposure (Fig. 4, left; Table 2; Supplementary Table 4).

Using this framework, EBV exposure was associated with a pronounced inflammatory signature in paediatric LC (Fig. 4, left; Table 2; Supplementary Table 4). EBV-experienced patients showed an innate-inflammatory cytokine profile with elevated IL-1α and IL-15, accompanied by higher IFNα and IL-18 (Table 2; Supplementary Table 4). EBV exposure was further linked to a Th17/Th22-associated pattern characterised by increased IL-12p40 and IL-22, whereas IL-17A/F did not show EBV-dependent differences after Holm–Bonferroni adjustment (Table 2; Supplementary Table 4). This was paralleled by increased regulatory cytokines (IL-11, IL-10 and IL-27), consistent with a concomitant counter-regulatory response (Table 2; Supplementary Table 4). Cellular innate activation was supported by higher neutrophil counts, with no increase in eosinophils or basophils (Table 2; Supplementary Table 4); across EBV-associated readouts, TsinceIndex, age and sex frequently contributed as covariates, whereas comorbidity did not, and vaccination status contributed only to neutrophil variation

## Table 2 | Associations of EBV exposure status with cytokines, granulocytes, and autoantibody levels in paediatric LC

| Variable | a. Anti-SARS-response | | | | b. SARS-related cytokines | | | | c. Innate-associated cytokines | | | | |
|---|---|---|---|---|---|---|---|---|---|---|---|---|---|
| | Anti-Spike | CH50U | C3 | C4 | IFNα | IL-13 | IL-6 | IL-1β | IL-1α | GM-CSF | TNFα | IL-15 | IL-18 |
| $R^2m$ | 0.1561 | 0.0521 | 0.0411 | 0.0237 | 0.1839 | 0.1548 | 0.1284 | 0.1055 | 0.1466 | 0.0894 | 0.1282 | 0.1951 | 0.1406 |
| $P$ | 0.0093 | 0.5401 | 0.7451 | 0.8734 | <0.0001 | <0.0001 | 0.0030 | 0.0865 | 0.0044 | 0.1253 | 0.0327 | 0.0003 | 0.0092 |
| Adj. $P$ | **0.0372** | 1.0000 | 1.0000 | 1.0000 | **0.0003** | **0.0003** | **0.0030** | 0.1730 | **0.0220** | 0.1730 | 0.0981 | **0.0018** | **0.0368** |

| Variable | d. Th1/2 ratio | e. Th17/22-associated cytokines | | | | f. Regulatory cytokines | | | g. Autoantibodies | | | | |
|---|---|---|---|---|---|---|---|---|---|---|---|---|---|
| | IL-4/IFNγ | IL-12p40 | IL-17A | IL-17F | IL-22 | IL-11 | IL-10 | IL-27 | Anti-ProT | Anti-CCP | Anti-TransG | M3-mAChr | M4-mAChr |
| $R^2m$ | 0.0251 | 0.1641 | 0.0619 | 0.1014 | 0.1354 | 0.1648 | 0.1280 | 0.0993 | 0.1582 | 0.1262 | 0.0571 | 0.1124 | 0.0877 |
| $P$ | 0.8741 | 0.0080 | 0.3892 | 0.0561 | 0.0056 | <0.0001 | 0.0160 | 0.0051 | 0.0002 | 0.0690 | 0.3208 | 0.0313 | 0.0858 |
| Adj. $P$ | 0.8741 | **0.02240** | 0.3892 | 0.1122 | **0.0224** | **0.0003** | **0.0160** | **0.0102** | 0.0010 | 0.2070 | 0.3208 | 0.1252 | 0.2070 |

| Variable | h. Granulocytes* | | | i. Mental health | | |
|---|---|---|---|---|---|---|
| | Neutrophils | Eosinophils | Basophils | SF12 PCS | PHQ-9 | GAD-7 |
| $R^2m$ | 0.2139 | 0.1244 | 0.0622 | 0.0825 | 0.1127 | 0.0730 |
| $P$ | 0.0001 | 0.0365 | 0.3863 | 0.2156 | 0.0231 | 0.1795 |
| Adj. $P$ | **0.0003** | 0.0730 | 0.3863 | 0.3590 | 0.0693 | 0.3590 |

Two-sided linear mixed models (LMMs) (Fig. 4, left; Supplementary Table 4) were used to assess associations between the indicated dependent variables and EBV exposure status in paediatric LC patients, accounting for repeated measurements clustered by patient ID. EBV exposure was modelled as a binary factor, distinguishing EBV-naïve patients (anti-EBV EBNA < 50 U mL⁻¹) from EBV-experienced patients (anti-EBV EBNA > 50 U mL⁻¹). Models included sex, vaccination status, and comorbidity as categorical factors, and TsinceIndex and age as covariates. Test statistics, estimates, 95% confidence intervals, degrees of freedom, and exact $P$ values for fixed effects are provided in Supplementary Table 4. Holm–Bonferroni-adjusted $P$ values (Adj. $P$) are shown. Bold numbers indicate dependent variables for which the overall LMM remained significant after Holm–Bonferroni adjustment and for which the estimated association with EBV exposure was positive.
*Anti-Spike anti-Spike SARS-CoV-2 ab, Anti-TransG anti-Transglutaminase, Anti-ProT anti-Prothrombin aAb.*

### Table 3 | Three of 43 clinical metabolic and coagulation parameters remain significant after mixed-effects modelling and multiple-testing adjustment using the Holm–Bonferroni method

| Parameter | $P$ | $R^2m$ | Adj. $P$ |
|---|---|---|---|
| TSH | <0.0001 | 0.4141 | **0.0043** |
| aPTT | <0.0001 | 0.2477 | **0.0043** |
| Lp(a) | 0.0001 | 0.1816 | **0.0123** |
| Vitamin D | 0.0039 | 0.1721 | 0.1482 |
| Zinc | 0.0545 | 0.1071 | 1.0000 |

Two-sided linear mixed-effects models (LMMs) were used to assess associations between the indicated factors and 43 blood-based clinical laboratory parameters in paediatric LC, with participant ID included as a random intercept. TsinceIndex (≤1 year vs 1–3 years), EBV exposure (anti-EBV EBNA < 50 U mL⁻¹ vs >50 U mL⁻¹), and DFS70 autoantibody status (positive vs negative) were modelled as binary fixed effects, and all two-way and three-way interactions among these factors were included. Shown are selected clinical parameters, including the three that remained significant after Holm–Bonferroni adjustment for multiple testing across all 43 parameters. Test statistics, estimates, 95% confidence intervals, degrees of freedom, and exact $P$ values for the significant models and effect-level summaries of all 43 parameters are provided in Supplementary Tables 5a and 5b.
*TSH thyroid-stimulating hormone, Lp(a) lipoprotein(a), aPTT activated partial thromboplastin time.*
Bold values indicate statistical significance of adjusted (Adj.) $P ≤ 0.05$.

(Fig. 4, left; Supplementary Table 4). In contrast, SARS-CoV-2-specific humoral readouts and complement measures, as well as aAbs, did not show EBV-dependent effects in this cohort/TsinceIndex window (Table 2; Supplementary Table 4). As chronic EBV responses are known to increase the risk of depression in adolescents[37], and depression and anxiety negatively impact recovery in children with chronic fatigue syndrome[38], the model was applied to mental health scores (Table 2; Supplementary Table 4). EBV serostatus was not associated with worse mental health scores after Holm–Bonferroni adjustment (Table 2). To synthesise these findings, we generated a composite overview integrating readouts with a significant overall model fit and a significant positive contribution of EBV exposure (Fig. 4, left; Table 2; Supplementary Table 4). This composite highlights a predominantly innate-inflammatory and IL-12p40/IL-22-skewed cytokine landscape in EBV-experienced paediatric LC patients, accompanied by increased neutrophils and regulatory cytokines.

### Verification of subgroups of paediatric LC

We identified three features by which LC subgroups can be categorised: The first relates to TsinceIndex, where LC in the first year is characterised by Th2-, innate and viral-associated cytokines, and LC persisting for 1–3.2 years is characterised by Th2-biased, Th17-related and innate-like systemic cytokines. A second relates to anti-DFS70-positivity, which was associated with markedly fewer coagulation abnormalities. The third involves EBV serostatus, which was linked to a pro-inflammatory cytokine profile and granulocytic dysregulation. To further explore these separately immuno-clinical subgroups in paediatric LC, we systematically analysed a panel of 43 haematopoietic, coagulation, electrolyte, and vitamin-related biomarkers (Table 3, Supplementary Table 5a, b). Using LMMs for each parameter, we tested for associations with subgroup assignment, EBV exposure status, anti-DFS70 autoantibody status, and time window based on TsinceIndex (<1 year vs 1–3 years), accounting for repeated measures by including participant ID as a random intercept. The 43 parameters assessed included thyroid-stimulating hormone (TSH), activated partial thromboplastin time (aPTT), lipoprotein(a) [LP(a)], electrolytes (Na⁺, K⁺, Cl⁻, Ca²⁺, bicarbonate), vitamins (B1, B6, B12, D, folic acid), metabolic markers (glucose, lactate, creatinine, eGFR), and inflammation/coagulation markers (CRP, D-dimer, fibrinogen, Factor VIII, protein S). After correcting for multiple testing, three metabolic markers were found to be significantly associated with the defined subgroups: TSH (adjusted $P = 0.0043$), aPTT (adjusted $P = 0.0043$)

and LP(a) (adjusted $P = 0.0123$) (Table 3). LP(a) and TSH levels increased over time, whereas aPTT decreased (Supplementary Table 5b). Significant models were adjusted for age and sex (Supplementary Table 5b). In the aPTT model, male sex was associated with higher aPTT values and improved the $R^2m$ by 0.084. In the LP(a) model, sex was statistically significant but contributed little to model fit, whereas sex was not significant in the TSH model. Age was not a significant predictor in any model. These biomarkers exhibited non-overlapping association patterns, supporting the concept of pathophysiological divergence. Together, these findings provide evidence for biologically differentiated subgroup structure within paediatric LC, although discrete endotypes remain to be established.

### EBV-naïve/low subgroup reveals immunometabolic correlates of functional status

Leveraging the anti-EBV EBNA–negative/low status observed in ~50% of our paediatric LC cohort, we applied LMMs used for disease severity in Fig. 2k (Bell score as dependent variable) to this subgroup, omitting EBV status as a covariate and retaining IL-12p40, MCHC, basophil counts and vitamin B1 (thiamine) as predictors (Supplementary Table 6). The model was highly significant ($P < 0.0001$) and showed an improved $R^2m$ of 0.3180 (Fig. 4, right; Supplementary Table 6a), supporting an immunometabolic contribution to clinical severity within EBV-naïve/low participants. IL-12p40 levels and basophil counts were the strongest positive predictors of Bell score, whereas higher MCHC was associated with lower Bell scores, consistent with poorer functional status at higher MCHC. Adding potential confounders (comorbidity, vaccination, sex, age and TsinceIndex) did not materially improve the model fit ($R^2m = 0.3169$; $P < 0.0001$; Supplementary Table 6b). Together, these data identify an EBV-naïve/low subgroup within paediatric LC with severity-linked immunometabolic and haematological features, including higher IL-12p40, basophil counts, vitamin B1 and anti-DFS70 positivity in participants with better functional status.

## Discussion

Here, we identify time-structured immune and metabolic dysregulation as a prominent feature of paediatric LC and describe biologically coherent subgroups linked to milder disease. These findings suggest clinically relevant immune heterogeneity in paediatric long COVID. Our data further indicate that paediatric LC may differ from adult LC. We found no consistent evidence for overt cardiac injury, systemic autoimmunity, complement activation, or EBV reactivation as dominant drivers of the observed post-acute phenotype[31,39–42]. This relative sparing is consistent with paediatric physiology and immune features, including higher regenerative capacity, greater thymic output and a predominantly naïve and diverse T-cell repertoire, as well as a lower cumulative inflammatory and antigenic burden and greater developmental immune plasticity, resulting in less rigid immunological imprinting.

A key interpretive point is that these three axes of heterogeneity are not mutually exclusive 'patient types', but partially independent programmes that can co-occur and change over time. The axes capture (i) a waning early antiviral/convalescent signature, (ii) a persistent innate-cytokine inflammatory programme, and (iii) a recovery-associated axis marked by immune–haematologic features linked to lower symptom burden. This framework links cross-sectional contrasts to longitudinal trajectories and argues against a simple linear inflammation–severity relationship.

Paediatric immunity is developmentally tuned across childhood and adolescence and is further modulated by puberty-associated endocrine changes. Consistent with this, our data show a persistent Th2-leaning and innate-cytokine programme beyond the acute phase alongside a waning early antiviral axis, supporting biologically heterogeneous post-infectious trajectories in established paediatric LC. A

central contribution of our work is its longitudinal depth: repeated assessments over up to 3 years across ~200 parameters reveal temporally dynamic immune imprinting rather than static post-infectious abnormalities. Innate- and Th17-associated signatures with features of chronic low-grade inflammation evolved over time, and patients followed heterogeneous trajectories (improving vs worsening), consistent with concurrent protective and pathogenic immune arms whose relative dominance shifts longitudinally[31,40]. This longitudinal structure also helps explain why adult-derived frameworks may not fully recapitulate paediatric biology: in adults, cytokine and aAb patterns often resemble EBV-positive cohorts[43,44], whereas in children, the lower prevalence of latent EBV infection may preserve discriminatory power to detect cytokine and metabolic correlates of reduced fatigue and recovery-associated trajectories.

The association of better lung function (higher FEV1 $z$-scores) with elevated systemic IL-13 suggests that repair-oriented programmes can operate in parallel with pro-inflammatory mechanisms, consistent with IL-13-linked epithelial repair following viral injury[29,45]. These findings underscore the importance of considering subclinical immunological alterations in paediatric LC cohorts, as even changes within normal limits may still have long-term implications for respiratory health[46]. Likewise, NfL percentiles correlated strongly with Bell scores despite values largely remaining within reference limits: severe cases preferentially occupied the upper tail of the normal distribution, while only children with higher functional status consistently showed NfL values in the lower normative range. Together, these distributional shifts support the concept that 'within-normal' biomarkers can still capture biologically meaningful organ involvement in paediatric LC.

Cytokines implicated in antiviral and anti-SARS-CoV-2 immunity (e.g. IL-13, IL-33 and IFNα) were markedly elevated during the first year[29,31,47] reflecting a waning SARS-CoV-2-associated immune signature consistent with a temporally structured immune trajectory. Although these cytokine alterations remained within physiologically tolerable ranges, their consistent association with disease phase suggests potential utility as candidate biomarkers for staging paediatric long COVID and monitoring longitudinal trajectories. In contrast, paediatric LC remained persistently innate-cytokine–weighted across all time points, aligning with evidence that innate programmes are comparatively prominent in paediatric antiviral responses[1,47]. Elevated innate cytokines, as observed here, may not only sustain inflammation but also contribute to symptom domains such as mood alterations[48]. Notably, paediatric LC exhibits a persistent cytokine dysregulation distinct from adult LC, with a Th2/Th1 shift towards Th2 that persists beyond 1 year[5]. Although a Th2-leaning immune bias has been proposed in children and female adolescence more broadly, we observed no age- or sex-dependence in our cohort (LMM, $P = 0.8$). To our knowledge, the combination of sustained Th2 skewing and persistently elevated innate cytokines has not been reported after mild SARS-CoV-2 infection or in most other viral convalescents[29].

When stratifying paediatric LC by EBV exposure, we observed a selective association of EBV-experienced status with components of the antiviral, innate and regulatory cytokine network, but not with a broad autoantibody signature. These data argue that EBV infection in paediatric LC is linked to a more activated antiviral/innate cytokine milieu and a neutrophil response, rather than to a generalised break in B-cell tolerance or a distinct aAb-defined subgroup. In postural orthostatic tachycardia syndrome, which can also occur post SARS-CoV-2 infection, GPCR aAbs have been similarly elevated as reported here, with levels not differing from controls and no correlation with disease severity[49]. We did not observe a consistent stratification of the three axes of heterogeneity by age or sex, and adjusting for both did not materially change the associations. Importantly, EBV reactivation appeared rare in line with EBV reactivation known from other paediatric cohorts, contrasting adult LC where reactivation can define a subgroup; in children, EBV serostatus thus acts less as a transient

epiphenomenon of viral stress and more as a stable imprint of prior herpesvirus experience that segregates a neutrophil–cytokine–biased subgroup. This EBV-linked subgroup was further supported when extending the analysis to 43 routine blood parameters. This constellation mirrors innate/inflammatory endotypes described in paediatric sepsis/influenza contexts where EBV exposure can associate with more complicated courses, supporting EBV serostatus as a biologically meaningful classifier in paediatric LC. If confirmed in independent cohorts, EBV serostatus could therefore represent a clinically accessible stratification variable for interpreting inflammatory signatures or designing subgroup-informed studies in paediatric long COVID. While EBV seropositivity and innate activation are also seen in MIS-C, the chronic immunometabolic profile observed here argues for a protracted and mechanistically distinct process compared with the acute, hyperinflammatory MIS-C phenotype[37,50].

We also identify features linked to a more favourable course, several of which are routinely measurable in clinical laboratories and may complement symptom-based assessment of disease course or recovery trajectories in paediatric LC. A decrease in MCHC with improving Bell scores may reflect shifts in microcirculatory dynamics; although MCHC is not a direct viscosity readout, reductions could indicate improved erythrocyte hydration or membrane stability and thereby oxygen delivery[32,51,52]. Given reported erythrocyte deformability defects in adult LC, recovery-associated shifts in paediatric red-cell rheology may represent a tractable component of resolution[32]. IL-12p40, mainly known as a subunit shared by IL-12 and IL-23, can also bind to the IL-12 receptor as a monomeric antagonist[53]. As IL-12p40 is typically present at concentrations far exceeding IL-23, it may dampen IL-12 signalling by acting as an IL-12 receptor antagonist, thereby limiting inflammation, consistent with its association with milder disease. Although IL-23 remained prolonged, adolescent post-infective chronic fatigue (SI-CFS) data show higher IL-23 in convalescent than active patients 24 months post infection, supporting our interpretation that persistent IL-23/IL12p40 may mark a protective, pro-resolution (healing) axis rather than ongoing pathology[54].

Participants with lower symptom burden (Bell score >40) tended to show NfL values in the lower normative range, whereas those with higher symptom burden more often fell into the upper tail of age-referenced norms. While compatible with biological heterogeneity in neuro-axonal stress, circulating NfL is not disease-specific, and elevations within (or near) the normative upper tail do not establish neuronal involvement. This pattern differs from what is typically observed in MIS-C[55], suggesting potentially distinct underlying drivers, but head-to-head studies are needed to test this and to assess its prognostic value for trajectories or long-term neurological outcomes.

In contrast, baseline factors that shape infection and LC incidence were of limited value for stratifying established paediatric LC. In our cohort, neither vaccination status nor anti-SARS-CoV-2 spike antibody titres were associated with symptom burden or immune-metabolic subgroups. This is consistent with population-based analyses[56]. Similarly, comorbidity, age, sex, viral variant, and number of infections prior to the index infection showed no robust associations with symptom burden or immunological profiles within this case-only cohort, limiting their usefulness for subgroup-based stratification.

Our study has several key strengths, including, to our knowledge, one of the most deeply phenotyped paediatric LC cohorts reported to date, integrating extensive clinical assessment with broad immune-metabolic profiling. Methodologically, the longitudinal design combined with LMMs enabled robust estimation of group differences while accounting for within-individual correlation, unbalanced sampling, and heterogeneous TsinceIndex. Limitations include restricted comparator biospecimen availability (limited to cytokine and aAb assays); therefore, additional readouts were analysed within the LC cohort and benchmarked against references with multiple-testing control. As an observational study, associations require cautious interpretation and

replication in independent cohorts. We also did not systematically capture social determinants of health (for example, food insecurity), which should be incorporated in future studies of LC severity[57].

Our data also refine the interpretation of cardiovascular risk signals reported in large electronic health record-based studies. Our deeply phenotyped cohort with long-standing LC showed no excess of manifest cardiac involvement: only three of 74 displayed abnormalities on echocardiography, consistent with background frequencies, while RECOVER analyses reported increases in coded cardiovascular signs/symptoms/diagnoses within weeks after infection in children and adolescents[58]. Differences likely reflect ascertainment (coded diagnoses versus detailed clinical work-up), disease phase (early post-acute vs established multi-year LC), and power for rare events, with our analyses focused on cardiac structure and function rather than vascular parameters or autonomic dysregulation. Thus, once paediatric LC is established, overt structural/functional heart disease does not appear to define a dominant or clinically informative subgroup in our setting. These negative findings also help delimit priorities for future study.

Finally, immune dynamics were interwoven with haematological, coagulation, metabolic and endocrine perturbations. Interaction structures across routine parameters were consistent with discrete subgroup structure in paediatric LC rather than a single severity continuum: the TsinceIndex window × anti-DFS70 interaction suggests time-dependent links to aPTT, and the anti-DFS70 × EBV interaction suggests a combined coagulation phenotype not explained by additive effects. Such non-additive, context-dependent biology converges with immuno-clinical clustering and underscores paediatric LC as a heterogeneous, mechanistically layered condition that may contribute to the limited success of prior 'one-size-fits-all' approaches.

Taken together, these findings suggest that the identified immune and clinical axes may represent a biologically informed framework for paediatric long COVID. If validated in independent cohorts, this framework could help define candidate endotypes and support risk stratification, longitudinal monitoring, and the design of subgroup-informed observational and interventional studies by distinguishing pathways linked to persistent symptom burden from those associated with recovery or resilience. Importantly, several candidate markers emerging from these axes are routinely accessible in clinical practice, potentially facilitating future translation into pragmatic prognostic and monitoring strategies.

## Methods
### Human participants and recruitment
Children and adolescents with suspected post-COVID-19 condition were recruited through the dedicated Long-COVID outpatient clinic at the University of Jena. Referrals were made by board-certified paediatricians who identified patients with a clinical suspicion of LC. During the recruitment period, school-aged children in Germany underwent routine screening with rapid antigen tests at least twice weekly; positive results were confirmed by RT-PCR, enabling precise determination of the index infection date for all included participants. At enrolment, a structured history explicitly captured any pre-existing symptoms and comorbidities prior to SARS-CoV-2 infection (Supplementary Table 1; Supplementary Table 7 for comorbidities in the paediatric population). Eligibility required that symptoms used for case definition first appeared after the confirmed infection, interfered with daily activities, and that both the patient and a parent/guardian consented to study participation. Case definition followed the National Institute for Health and Care Excellence (NICE) rapid guideline on long-term effects of COVID-19 (published December 2020; updated November 2021).

Of 106 patients evaluated in the clinic, 78 met prespecified criteria and were proposed for inclusion. All proposed cases were independently reviewed by the study lead to verify case ascertainment; one

patient was excluded at this stage because relevant symptoms clearly pre-dated infection and could not be reliably distinguished from a post-COVID-19 condition. The final analysis set comprised 74 participants who completed full data collection at both study visits. Cohort characteristics and clinical manifestations (sex, age, vaccination status, comorbidities, VOC attribution, prior SARS-CoV-2 infections) are provided in Supplementary Table 1a, b and Fig. 1. The control group comprised healthy children and adolescents and paediatric cystic fibrosis (CF) patients in stable condition (Supplementary Table 1d). Controls were recruited independently and were slightly younger than the LC cohort (mean ± SD, 10.9 ± 5.15 vs 14.1 ± 2.5 years; Mann–Whitney $U$, $P = 0.07$; Supplementary Table 1a). Control samples were available for cytokine and aAb assays; other readouts relied on within-cohort stratification and/or external reference datasets where applicable. Robustness to control composition was assessed by comparing healthy vs CF controls and repeating key cytokine and autoantibody analyses as LC–healthy and LC–CF contrasts; conclusions were unchanged, supporting pooling for these assays. CF served as an infection-exposed chronic respiratory disease comparator and was not intended as an inflammation-free baseline.

## Patient assessments

Paediatric LC patient assessments comprised evaluation by a psychologist and a board-certified paediatric neurologist using standardised, age-appropriate neuropsychological instruments. The assessments comprised validated questionnaires and objective physical performance tests (Fig. 1a, e; Supplementary Fig. 1a). Most questionnaires for participants were completed by proxy. Psychometric and functional assessments included the Generalized Anxiety Disorder 9-item scale (GAD-7) for anxiety symptoms, the Patient Health Questionnaire 9-item (PHQ-9) for depressive symptoms, and the Short Form-12 Physical and Mental Component Summary scores (SF-12 PCS and SF-12 MCS) to evaluate physical and mental health-related quality of life. Vital parameters and a wide range of symptoms were assessed through these instruments, including the Munich Long COVID Symptom Questionnaire (MLCSQ), which assesses the frequency of 96 potential symptoms of LC, divided into 13 Systems (Fig. 1a, c; Supplementary Fig. 1). Symptom burden was captured through both parent-reported symptoms (PRS) and patient-reported symptoms (PtRS). Child-specific health-related quality of life in children was assessed using the KIDSCREEN questionnaire, and sleep disturbances were evaluated with the Children's Sleep Habits Questionnaire (CSHQ). Functional impairment was assessed using the Bell Disability Scale (Bell score), while fatigue was evaluated with the Fatigue Severity Scale (FSS).

## Cardiopulmonary and vascular assessment

All children and adolescents underwent a standardised cardiopulmonary assessment performed by paediatric cardiologists and pulmonologists. This included structured cardiac and respiratory histories addressing pre-existing disease, exertional symptoms (including chest pain, palpitations, syncope or presyncope, dyspnoea and cough), and relevant family history, as well as focused cardiovascular and pulmonary physical examinations. Resting blood pressure was measured using age-, sex- and height-adjusted reference standards. Baseline cardiac evaluation comprised a resting 12-lead electrocardiogram (ECG) and transthoracic echocardiography. ECGs were assessed for rhythm disturbances, conduction abnormalities and repolarisation changes. Echocardiography evaluated cardiac chamber dimensions, global systolic and diastolic function, valvular morphology, and congenital structural abnormalities. Pulmonary function testing included standardised spirometry performed in accordance with international paediatric guidelines, with forced expiratory volume in 1 s ($FEV_1$) and related parameters expressed as age-, sex- and height-adjusted $z$-scores. Pulmonary gas exchange was assessed by single-breath diffusing capacity for carbon monoxide (DLCO), adjusted for age, sex and body size. Fractional exhaled nitric oxide (FeNO) was measured as a non-invasive marker of airway inflammation using standardised procedures. Physical performance was quantified using the Sit-to-Stand Test (STS) and Hand Grip Strength Test (Fig. 1e, Supplementary Fig. 1)[59,60].

In cases where symptoms or clinical findings suggested further systemic organ involvement, patients were evaluated by the respective paediatric subspecialist, and further diagnostic work-up was performed as clinically indicated.

## Biospecimen processing and laboratory assays

Peripheral blood was collected into EDTA and SST tubes. Serum was analysed immediately or stored at −20 °C for batched cytokine and aAb assays, using once thawing only. PBMCs were isolated by Ficoll-Paque and cryopreserved for biobanking (PBMC-derived assays were not included in this manuscript).

## Cytokine analysis in serum

Cytokine analyses were conducted as pre-specified primary immune-axis analyses using pre-defined functional families with a priori lead markers. Cytokine levels in serum samples were quantified using a flow cytometric multiplex bead-based assay (Supplementary Fig. 5)[20,31]. To assess robustness to control composition, we repeated the cytokine analyses without pooling controls by evaluating healthy controls and CF controls separately versus LC using the same statistical workflow. Conclusions were unchanged, supporting pooling of control groups for these assays. Data acquisition was performed on a FACSFortessa X-20 (Becton Dickinson), and results were analysed using the LegendDPlex Data Analysis Software Suite (Qognit). Cytokines were organised into pre-specified functional families (A–F). For each cytokine, group differences were assessed using a Kruskal–Wallis test followed by Dunn's post hoc tests for pre-specified pairwise comparisons. Dunn $P$ values were adjusted within each cytokine family using the Holm–Bonferroni method. Adjusted $P$ values > 1.0 were truncated to 1.0; adjusted $P < 0.05$ were considered significant.

## Routine laboratory testing

Laboratory analyses included metabolic markers, blood gases, differential blood counts, general clinical chemistry, coagulation parameters, urinalysis, immunoglobulins, hormone levels, allergy diagnostics, autoantibodies, antiviral antibodies (EBV- and CMV-specific), and anti-SARS-CoV-2 serology. Pulmonary function parameters, vascular function (static and dynamic analysis), and vital signs were also assessed. Arterial/venous blood gases were analysed on ABL 90 Flex Plus instruments. Complete and differential blood counts were obtained on an XN-1000 (Sysmex) using fluorescence flow cytometry, SLS-Hb and impedance channels. EBV serostatus was determined by anti-EBV EBNA1 IgG (anti-EBV EBNA) and used to define prior EBV exposure (EBV-experienced: ≥50 U mL$^{-1}$; EBV-naïve: <50 U mL$^{-1}$). The threshold was pre-specified to separate background-level measurements from clearly elevated titres across the assay's dynamic range; anti-EBV EBNA was used as an exposure marker and was not interpreted as evidence of EBV reactivation. Anti-SARS-CoV-2 spike S1 IgG (quantitative; ELISA Genie, CBK4154) and anti-nucleocapsid IgG (qualitative; Roche Elecsys) were performed per manufacturer's instructions. Values exceeding analytical limits of detection were recorded as maximum + 1 or minimum/2, respectively.

## Autoantibody profiling and biomarker quantification

Antinuclear antibodies (ANA) were detected by indirect immunofluorescence using the ANA HEp-2 plus kit (#8101, Generic Assays, Dahlewitz, Germany) on HEp-2 cells following serial serum dilutions (1:80–1:2560; positivity defined as ≥1:80), with endpoint titres evaluated using a fluorescence microscope (Olympus, Japan). Presence of

anti-DFS70 (LEDGF/p75) autoantibodies (anti-DFS70) was confirmed by immunoblot using the EUROLINE Anti-DFS70 (IgG) kit (#DL 159z-1601 G, Euroimmun, Lübeck, Germany).

Anti-CCP IgG (#3665, cut off <30 U mL$^{-1}$), anti-TransG IgA (#4033, cut off <20 U mL$^{-1}$; both Generic Assays, Dahlewitz, Germany) as well as anti-Cardiolipin IgG (#ORG 515S, cut off <10 U mL$^{-1}$), anti-β2-Glycoprotein IgG (#ORG 521S, cut off <10 U mL$^{-1}$), anti-Prothrombin IgG (#ORG 541S, cut off <20 U mL$^{-1}$), anti-PR3 IgG (#ORG 618, cut off <10 U mL$^{-1}$) and anti-MPO IgG (#ORG 519, cut off <5 U mL$^{-1}$; all Orgentec, Mainz, Germany) were analysed by ELISA. Serum samples were diluted 1:100 in sample buffer and transferred in duplicate into the respective microtiter plate together with standards and controls. Final analysis was performed using a Tecan Sunrise Microplate reader (Tecan, Männedorf, Switzerland) and calculated according to the linearity scale. Values below the diagnostic thresholds were retained as described previously to illustrate the assay's linear range[29,31], acknowledging the primarily diagnostic role of established cut-offs. To assess robustness to control composition, we repeated key aAb analyses using a three-level group factor (LC, healthy controls, CF controls) and extracted LC–healthy and LC–CF contrasts. Conclusions were unchanged, supporting pooling of control groups for these assays.

GPCR autoantibodies. Serum IgG autoantibodies directed against β1- and β2-adrenergic receptors and M3- and M4-muscarinic acetylcholine receptors were quantified using a standardised commercial ELISA (CellTrend GmbH) according to the manufacturer's instructions and are reported in U mL$^{-1}$. Assay-defined reference ranges (manufacturer-recommended cut-offs) were <15 U mL$^{-1}$ for β1-adrenergic, <8 U mL$^{-1}$ for β2-adrenergic, <6 U mL$^{-1}$ for M3-muscarinic, and <10.7 U mL$^{-1}$ for M4-muscarinic receptor aAbs; the upper assay limit was > 40 U mL$^{-1}$. This assay measures aAb binding and does not directly assess agonistic or antagonistic activity; therefore, results are interpreted as binding aAb readouts rather than a direct functional measure.

Analysis of serum neurofilament light chain (NfL) levels was conducted using the Quanterix Simoa NF-Light assay Advantage Kit (Lexington, MA), according to the protocol provided. All samples were analysed in duplicate. Testing was performed blinded to patient clinical/paraclinical data and outcome measures. To adjust for the influence of age, NfL concentrations were converted to Z-scores and (interchangeable) percentiles https://shiny.dkfbasel.ch/baselnflreference-for-kids[35]. The z-score represents the number of standard deviations a particular value deviates from the mean of healthy, age-matched individuals[35].

## Statistical analysis
Analyses were organised around pre-specified primary objectives anchored in an a priori subgroup framework (TsinceIndex, EBV exposure, and an autoantibody axis), complemented by exploratory analyses for hypothesis generation. Primary inference focused on these pre-defined axes and their specified interactions; broader biomarker screens were treated as exploratory and interpreted accordingly. Full model specifications and outputs are provided in the Supplementary Tables.

**Definitions.** Time since the LC-initiating ('index') SARS-CoV-2 infection (TsinceIndex) was defined as the interval between the last documented SARS-CoV-2 infection preceding LC symptom onset and the study visit. Reinfections occurring after LC onset were treated separately and did not redefine the index infection. TsinceIndex was analysed either continuously (weeks) or categorised as ≤52 weeks versus >52 weeks up to 166 weeks; for readability, these are reported as <1 year and 1–3 year (i.e. up to 3.2 years). EBV exposure was defined using anti-EBV EBNA titres (cut-off 50 U mL$^{-1}$). The aAb axis was operationalised using anti-DFS70 status (positive/negative).

**Handling of repeated visits.** Participants could contribute up to two study visits. Baseline demographics and symptom measures collected at enrolment were analysed using visit 1 only. For cross-sectional LC-versus-control comparisons within a given time window, one value per participant per window was used; if two assessments occurred within the same window, values were collapsed to the participant-level mean. Analyses explicitly addressing within-LC longitudinal change (e.g. early vs later LC) used mixed-effects modelling to account for repeated measures.

**Linear mixed-effects modelling.** LMMs were used for longitudinal and within-LC analyses, including a random intercept for participant ID (1|ID). Fixed effects were defined by the biological question and included, where appropriate, TsinceIndex (continuous or windowed), EBV exposure, anti-DFS70 status, age, sex, BMI percentile, and pre-specified clinical/exposure covariates (comorbidity, VOC wave, infections prior to LC onset, vaccination). Interaction terms were included only in models designed a priori to test effect modification (e.g. trajectory- or time-window-dependent associations).

**Between-group cytokine and aAb analysis.** Primary between-group contrasts for systemic cytokines and aAbs were presented as parsimonious LC–control comparisons aligned across time windows. Covariate-adjusted versions were performed as sensitivity analyses using clinically plausible covariates (comorbidity status, infections prior to LC onset, vaccination, TsinceIndex, age, and sex); conclusions were unchanged.

**Model diagnostics, inference, and multiple testing.** Model assumptions were assessed by inspection of residuals and Q–Q plots; multicollinearity was evaluated using variance inflation factors. Normality of variables used in non-model-based analyses was assessed using Shapiro–Wilk and Kolmogorov–Smirnov tests alongside Q–Q plots. P values for fixed effects were obtained using Satterthwaite's approximation and validated by bootstrap resampling (1000 iterations). Where multiple hypotheses were tested within a pre-defined test family, P values were adjusted using the Holm–Bonferroni method; adjusted $P < 0.05$ was considered significant.

**Additional tests and software.** Non-model-based group comparisons were performed using Kruskal–Wallis tests with Dunn's post-hoc tests for multiple-group comparisons and Mann–Whitney U-tests for two-group comparisons, as indicated. Correlations were assessed using Pearson's correlation for approximately normally distributed variables and Spearman's rank correlation otherwise; where repeated visits were present, associations were evaluated using two-sided LMMs with participant ID as a random intercept. Values below the assay detection limit were imputed as half the detection limit; values above the assay range were set to the upper limit + 1. The software used in this study included GraphPad Prism (version 10.6.1 (892)), Jamovi (version 2.7.12) with GAMLj (version 3), Adobe Illustrator (2025, Adobe Inc.), RStudio (version 2025.05.01+513, Posit PBC), FlowJo (version 10.10.0), and BioRender (BioRender.com; web-based application; accessed in April 2026).

## Ethics
Written informed consent was obtained from all participants and/or their legal guardians in accordance with the Declaration of Helsinki. The study was approved by the ethics committees of the University Hospital Jena (2022-2614_1-BO) and Otto-von-Guericke University Magdeburg (164-18). Although observational and non-interventional, the study is registered in the German Clinical Trials Register (DRKS00028523).

## Reporting summary
Further information on research design is available in the Nature Portfolio Reporting Summary linked to this article.

## Data availability

The data supporting the findings of this study are not publicly available because access is restricted by the study ethics approvals, participant informed consent, and the data governance regulations of the Long-COCID consortium. Requests for pseudonymized data for scientific research purposes may be submitted to Kinderklinik@med.uni-jena.de and will be reviewed by the governing board of the LongCOCID consortium in accordance with the consortium's governance procedures and applicable data protection regulations. The governing board aims to respond to data access requests within 8 weeks. If approved, access will be granted under a data use agreement and will be limited to research purposes consistent with the original ethics approvals, participant consent, and applicable data protection requirements.

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

## Acknowledgment

D.V. discloses support for the research of this work from the Multicentre long COVID registry (MLC-R) and from the BMBF (LongCOCID 01EP2101). M.C.B.W. discloses support from the BMBF (01EP2101C), the DFG (Br1860/18), and the Ministry of Science, Technology and Environment of Saxony-Anhalt (SarsImmunGender I-196).

## Author contributions

M.C.B.W., K.V., I.H., P.J., S.W., D.R., A.R., J.M., M.L., L.N., P.H. and E.U. performed or supervised experiments and generated and analysed data. E.U., D.V., M.P., C.A., H.P. and L.N. evaluated and recruited patients and/or controls. S.W. and M.C.B.W. applied, verified, and visualised the statistical methods. M.C.B.W. wrote the original draft with input from all co-authors and is the lead corresponding author. M.C.B.W., K.V., I.H., P.J., J.K. and D.V. performed computational analysis of data. D.V. and M.C.B.W. conceptualised the project. M.C.B.W. and D.V. directed the project and acquired funding.

## Funding

## Competing interests

The authors declare no competing interests.
