## [Peer Review file · Nature Communications]

Immune-metabolic trajectories delineate subgroups in paediatric long COVID

Corresponding Author: Professor Monika Brunner-Weinzierl

Version 0:

Reviewer comments:

Reviewer #1

(Remarks to the Author)

This manuscript by Brunner-Weinzierl investigated immunological and metabolic parameters in a cohort of paediatric long COVID, reported differences in some of these parameters associated with duration of long COVID, as well as differences in subgroups, such as those seropositive for Epstein-Barr virus. A key strength of this study is the breadth of molecular and clinical parameters investigated. However, the rationale of several aspects of the analysis are unclear. The main concerns are:

1. The diagnosis of LC in children remains somewhat debated. A key issue is whether children had symptoms prior to SARS-CoV-2 infection. Although the authors excluded one potential LC participant due to pre-COVID symptoms, it is not clear whether and how this exclusion criterion was sought in all participants
2. The sample size is relatively modest, and the authors have undertaken a wide range of immunological and metabolic investigations. The rationale for some of these is not well articulated. It is not clear beyond high level immune and metabolic dysregulation what the specific a priori research questions were. It would be informative for the introduction to draw on the adult (and more limited paediatric) literature to frame the specific hypotheses.
3. The methods describes PBMC separation, but no PBMC-related data are obviously reported, which is a pity; it would be interesting to know whether PBMCs and innate cells in particular show evidence of training, as has been reported.
4. It is unclear whether the LC followed the participants first COVID infection (given most were infected in the third wave) and whether any had been vaccinated
5. The choice of controls is interesting. The controls are CF patients and healthy children (their COVID exposure is not discussed). Both control groups are relatively small and they are combined for some analyses. The use of CF controls is justified by the authors "as they are frequently exposed to respiratory infections despite stable treatment and similarly affected by SARS-CoV-2 infection. This ensures that the immune changes observed in long COVID 92 are not driven by pre-existing inflammation or infection frequency, but rather reflect LC disease-specific mechanisms". The latter statement is quite a stretch, as CF patients are colonised with a very abnormal respiratory (and GIT microbiome) and have a markedly increased burden of predominantly bacterial exacerbations.
6. There are a lot of assays performed and it is somewhat difficult to ascertain whether the nominally significant results reflect a priori hypotheses
7. The authors combined the cystic fibrosis and the non-cystic fibrosis controls into a single control group on the basis of not observing markedly different IL-6 or TNF-alpha between these two subgroups. However, basing this on just two cytokines would appear to be overlooking potential differences between these two subgroups in many of the other immunological parameters investigated in this study. In addition, combining these two subgroups would seem to negate the stated aim of

recruiting cystic fibrosis controls to try and distinguish long COVID-specific effects from pre-existing high infection and inflammation burden.

8. Throughout, there are aspects of the analysis rationale that are unclear. The study would benefit from more consistent approach to analyses. Some specific issues:

i) Why were only three biomarkers selected for the investigating associations with lung function, when the broader range of biomarkers was used for other clinical measures? The authors highlight the relevance of IL-13 and IL-33 to lung function, but this approach seems to be quite limited.

ii) Analyses are presented for testing differences between the control group and the LC <1 year group, and investigating differences between the LC <1 yr and LC 1-3 yr group, but not between the control and LC 1-3 yr groups. This seems like a notable omission.

iii) There are inconsistencies in covariates included in models (notably, sex is adjusted for in some sets of models but not others), it would be helpful for authors to more clearly articulate how they selected their model and the differences. Another inconsistency is how data from participants with multiple visits were handled, with some analyses using the mean of these visits (e.g., Figure 2), others using a 'single representative value' (e.g., Figure 1) and others treating each observation as independent despite this approach biasing the presented analysis (e.g., the Spearman correlations in Extended Data Table 2.1c).

9. The value of the k-means approach used to define clusters of FEV1 z-scores is not well-articulated, with the decision to define three clusters appearing largely arbitrary. Could the authors please provide additional information on this approach including the metrics they used to validate their approach?

10. The lung function analyses do not appear to be discussed at all in the discussion section. The authors' discussion of these analyses would be beneficial.

11. The discussion would also benefit from clearer discussion of this study's limitations and strengths, which seem to have been omitted.

Some other more minor comments:

12. Introduction: "Autoimmune diseases were found to occur with greater frequency" – is this referring to autoimmune diseases as a risk factor of LC, or autoimmune diseases as a consequence of LC?

13. Results: "This indicates that different pathological mechanisms contribute to disease severity and progression." – is this overstating the causal evidence of these associations?

14. Results: why was IL-13 selected over IL-33 for the lung function models?

15. Table 1 + 2: What was the rationale for including interaction terms in the models presented in Table 2 but not the models presented in Table 1?

16. Methods: "The z-score represents the number of standard deviations from the mean of healthy age-matched controls" – this is largely duplicative text with the previous sentence.

17. Extended Data Figure 2 + 3.1: it is not clear what the different panels refer to.

18. EDT6 a + b – Why has information on fixed effects been provided only for the top models?

Reviewer #2

(Remarks to the Author)

Reviewer #3

(Remarks to the Author)

General comments:

In this article titled "Immune-Metabolic Programs Drive Disease Trajectories in Paediatric Long COVID", the authors describe aspects of both the clinical course and underlying biological differences in paediatric patients diagnosed with long COVID. Using a combination of survey data, blood analyte and cytokine profiling, as well as statistical analysis the authors describe finding differences both when comparing paediatric Long COVID patients to controls in addition to three different biologically distinct sub-categories of Long COVID. These findings are novel to the field. Paediatric Long COVID is an important area of study as there are significantly fewer studies evaluating the paediatric population compared to adults. As the authors point out, there appear to be different mechanisms involved, thus underscoring the need to study paediatric patients directly. Overall, the study design, the methods, and analysis are generally well constructed. My most significant criticism is that the rationale for many of the experiments done should be more clearly explained. There are many different hypotheses including Th2 vs Th17 immune skewing, EBV serostatus, neuro-axonal injury, and coagulopathies all with the added

complexity of timing of Long COVID illness. Clearly these are all important aspects to evaluate, but the study would be stronger if this was laid out in a more logical fashion for the reader to follow.

Figure 1/Demographics:

1. Including criteria of how Long COVID was diagnosed and how participants were selected: The methodology doesn't describe how the Long COVID patients were selected/diagnosed. As a clinical diagnosis, it would be helpful to understand how the clinicians determined if participants have Long COVID. Did they follow the WHO criteria? Where were they diagnosed and followed? Did families/children volunteer themselves?
2. Number of participants needed: How was 73 participants selected? Was a power calculation done? I see on the clinical trials website, the authors were looking to recruit 150 participants. How were the number of controls selected?
3. Age difference in control vs LC: The control group is younger and not age matched—which is something that might matter when discussing EBV related exposures.
4. Control group COVID-19 status: I am concerned that 61% of the control patients analyzed had not had SARS-COV-2. There is mounting evidence that any SARS-CoV-2 infection alters the immune system regardless of LC status. Do the authors findings still stand when the Long COVID group is compared to convalescent controls? This would be important to determine.
5. Chronic Disease prior to LC status: There are a number of LC patients who have other chronic illnesses. Should they be included in this analysis (ie: these diseases do not confound the data)? If they should be included, the rationale should be explained.

Clinical Features of Long COVID

1. FEV1 Z-score: It is interesting that the K modes analysis of FEV1 z-score showed 3 different groups. Was there any additional analysis done on these 3 groups (ie; in relation to blood analytes/cytokines)? If so, how did these compare to the control group?

Figures 2 and 3

1. Is there a reason that the LC>1 year and control plots are separated out from LC>1 year? Could they be all on the same plots? If there is a reason the data is separated out this way, it would benefit from further explanation.
2. The fact that statistical significance is represented after multiple corrections analysis should be more clearly stated in the figure legends. If those are the effects that have survived after correcting for 43 analytes, that is much more compelling.

Figure 4

This is a nice visual representation of a lot of data. It would be nice if there was a way to have some of the data from Table 1 as a separate panel in this figure also. Maybe even just a plot of EBV EBNA IgG level in the Long COVID group vs control.

Discussion:

In general, the discussion section would benefit from expanded rationale and thoughts regarding the different endotypes the authors describe. The comparison to adult LC is important however the manuscript would benefit from more explanation about their own findings. Are there age/sex contributions to each of those 3 types found? How does the development of the pediatric immune system progress over time—what is already known? What is known in relation to other chronic infections? How does EBV status shape immune system? There are some known age-related immune changes such as skewing from Th1 to Th2. How does this play into the authors' hypotheses and findings?

Version 1:

Reviewer comments:

Reviewer #1

(Remarks to the Author)

Thank you for the opportunity to review the revised submission.

The authors have undertaken considerable revision of this manuscript to address the previous reviews, and in doing so have clarified some of the concerns and details around the methodology of this study.

The sample size is modest and some of the large number of analyses are difficult to interpret with confidence.

The incorporation of additional analyses has better supported the approach taken to control groups in this study, although the use of CF patients introduces complexities in interpretation of the findings.

The authors have also indicated throughout where analyses were based on primary hypotheses or were exploratory/hypothesis-generating. However, even with these revisions, the messaging of the manuscript is made less clear by the broad range of hypothesis-driven and hypothesis-generating analyses across different sets of data and comparisons to investigate the various aspects of the authors' aims.

There are multiple immune and endotype 'axes' considered in this manuscript, and while the authors highlight specific associations of interest in the discussion from the milieu of analyses undertaken, the importance of these findings for the field or in the context of potential opportunities for prognosis or management are not sufficiently articulated.

Reviewer #2

(Remarks to the Author)

Reviewer #3

(Remarks to the Author)

This submission is significantly improved after the modifications the authors have made. Their rationale for study design and subsequent analysis is clear to follow.

The figures are also much improved and each have a clear message. My only small edit is that in table 1, I believe the second half with corrected p values should have one part that is C vs LC <1yr and C vs LC >1 yr (it currently has both as >1yr).

I thoroughly enjoyed reading the discussion and appreciate the idea of the three pediatric LC endotypes identified are mechanistic based and dynamic. This is an important concept for the field to consider.

I have no additional questions or concerns.

Immune-metabolic trajectories delineate endotypes in paediatric long COVID by Vilser et al and Brunner-Weinzierl,” for consideration at *Nature Communications*

Point-by-point reply

Reviewers' comments:

Reviewer #1 (Remarks to the Author):

Response: We thank the reviewer for the thorough and constructive evaluation. We have addressed the concerns by clarifying the a priori hypotheses and analysis rationale, strengthening signposting between experimental modules, and improving the presentation of controls and the pre-defined time windows across the main figures. Specific changes are detailed point-by-point below.

Reviewer #1: I co-reviewed this manuscript with one of the reviewers who provided the listed reports. This is part of the Nature Communications initiative to facilitate training in peer review and to provide appropriate recognition for Early Career Researchers who co-review manuscripts.

This manuscript by Brunner-Weinzierl investigated immunological and metabolic parameters in a cohort of paediatric long COVID, reported differences in some of these parameters associated with duration of long COVID, as well as differences in subgroups, such as those seropositive for Epstein-Barr virus. A key strength of this study is the breadth of molecular and clinical parameters investigated. However, the rationale of several aspects of the analysis are unclear.

Response: We thank the reviewer for the careful evaluation and agree that the rationale and structure of the analyses needed clearer signposting. We therefore made the a priori conceptual framework explicit in the Introduction (temporally dynamic immunometabolic endotypes; modulation by EBV exposure and autoreactivity; p. 4, l. 79–89). We also added/updated the study design overview (Fig. 1a) and now carry the control comparison together with the two pre-defined disease windows (<1 yr; 1–3 yr since index infection) consistently across Fig. 1d, Fig. 2 and Fig. 3 to provide a single backbone for the mechanistic modules (Fig. 1 legend p. 26, l. 657–666; Fig. 2 legend p. 28, l. 670–688; Fig. 3 legend p. 29, l. 694–708). In the Results, we now state which analyses were pre-specified (including the primary cytokine objective and the one-year window rationale; p. 7, l. 150–158) and we clarify the rationale for modelling EBV serostatus as a stable exposure/imprinting variable (p. 11, l. 268–276) and for the pre-specified severity model covariate set (p. 8, l. 195–212). Finally, we clarified definitions and handling of repeated visits/time windows in Methods (p. 37, l. 892–905).

We thank the reviewer for the thorough and constructive evaluation. We have addressed the main concerns by clarifying the a priori hypotheses and analysis rationale, strengthening signposting between experimental modules, and improving the presentation of controls and the pre-defined time windows across the main figures.

Reviewer #1: The main concerns are:

1. The diagnosis of LC in children remains somewhat debated. A key issue is whether children had symptoms prior to SAR-CoV-2 infection. Although the authors excluded one

potential LC participant due to pre-COVID symptoms, it is not clear whether and how this exclusion criterion was sought in all participants.

Response: We agree that paediatric LC case ascertainment critically depends on establishing symptom onset relative to SARS-CoV-2 infection. For all referred cases, enrolment included a structured history that explicitly captured pre-existing symptoms and comorbidities prior to SARS-CoV-2 infection, and eligibility required that symptoms used for the LC case definition first appeared after the confirmed index infection and interfered with daily activities. During the recruitment period, routine rapid antigen screening with PCR confirmation enabled precise determination of the index infection date. Of 106 patients evaluated, 78 met prespecified criteria and were proposed for inclusion; all proposed cases were independently re-reviewed by the study lead, and one candidate was excluded because relevant symptoms clearly predated infection. The final analysis set comprised 74 participants with complete data at both study visits. We have clarified these procedures in more detail in the Methods, now.

2. The sample size is relatively modest, and the authors have undertaken a wide range of immunological and metabolic investigations. The rationale for some of these is not well articulated. It is not clear beyond high level immune and metabolic dysregulation what the specific a priori research questions were. It would be informative for the introduction to draw on the adult (and more limited paediatric) literature to frame the specific hypotheses.

Response: We apologize for not being clear enough and have now stated more explicitly the a priori questions for motivating each analytical axis and better connect them to the adult and emerging paediatric literature. In the revised Introduction, we therefore articulate our central hypothesis that paediatric LC segregates into temporally dynamic immunometabolic endotypes in which waning SARS-CoV-2-associated antiviral/humoral signals interact with persistent innate- and Th2-skewed programmes and are further modulated by prior EBV exposure and patterns of autoreactivity (p. 4, l. 79–89).

We then specify the pre-defined questions and corresponding readouts throughout the Results and Methods: persistent/stage-dependent antiviral activity assessed a priori by serum IFN α (with complement parameters as contextual comparators) across the pre-defined time windows (p. 5, l. 102–106); temporally structured systemic cytokine perturbations tested as the pre-specified primary cytokine objective using pre-defined functional families with a priori lead cytokines and an explicit rationale for the one-year primary window (p. 7, l. 150–158); biological stratification by stable exposure/imprinting variables, operationalised a priori as EBV exposure and an autoantibody axis (anti-DFS70) (Methods p. 36, l. 840–844; p. 37, l. 895–898); and a targeted panel of routine haematological/coagulation/metabolic markers (43 routine laboratory parameters) to test whether standard clinical readouts recapitulate the same endotype structure (Abstract p. 2, l. 44–46; Discussion p. 16, l. 411–412).

Finally, we clarify which analyses are primary and hypothesis-driven versus exploratory/hypothesis-generating, including the corresponding multiple-testing strategy, and we note that lower-frequency signals are interpreted cautiously and intended to prioritise mechanisms for independent replication (p. 37, l. 885–890).

3. The methods describes PBMC separation, but no PBMC-related data are obviously reported, which is a pity; it would be interesting to know whether PBMCs and innate cells in particular show evidence of training, as has been reported.

Response: We agree that PBMC-based analyses from children—particularly functional and/or epigenetic assays to test trained immunity in innate cells—would be informative. One major concern was that we could not collect PBMCs from our healthy controls. LC-PBMC isolation was performed primarily for standardised biobanking for future mechanistic within the LC cohort studies rather than as a pre-specified readout reported here. In practice, rigorous trained-immunity workflows require dedicated experimental design (standardised re-stimulation and/or chromatin profiling) and sufficient viable monocyte yield. This was not achievable across our cohort because PBMCs were cryopreserved and, given the paediatric sampling constraints, cell numbers (and post-thaw monocyte recovery/viability) were limited in a substantial fraction of samples. Especially freeze–thaw is known to affect monocyte functional cytokine responses, reducing the robustness of TRIM inference from thawed PBMCs. We therefore did not include PBMC-based TRIM assays in the present manuscript and instead report extensive routine differential blood counts as broad in vivo context.

4. It is unclear whether the LC followed the participants first COVID infection (given most were infected in the third wave) and whether any had been vaccinated

Response: We agree this should be explicit. Prior SARS-CoV-2 infection history and vaccination status were systematically captured at enrolment and are summarised in Supplementary Table 1a. The LC-initiating “index infection” used for TsinceIndex was defined as the last documented SARS-CoV-2 infection preceding LC symptom onset (i.e., not necessarily the first infection; for participants with multiple infections, t₀ was the infection after which LC symptoms appeared, usually the last). In the LC cohort, 45/74 reported one infection prior to LC onset, 28/74 reported two, and 1/74 reported three (Supplementary Table 1a). Vaccination prior to LC onset was also recorded (0/1/2/3/4 doses: 37/9/21/6/1; Supplementary Table 1a). Number of vaccinations and number of infections prior to LC onset were included as pre-specified covariates in our mixed-effects models and were not associated with disease severity over the disease course; full model outputs are provided in Supplementary Table 1b/1d (lower panel) and referenced in the Results (p. 5, l. 112–119).

5. The choice of controls is interesting. The controls are CF patients and healthy children (their COVID exposure is not discussed). Both control groups are relatively small and they are combined for some analyses. The use of CF controls is justified by the authors “as they are frequently exposed to respiratory infections despite stable treatment and similarly affected by SARS-CoV-2 infection. This ensures that the immune changes observed in long COVID 92 are not driven by pre-existing inflammation or infection frequency, but rather reflect LC disease-specific mechanisms”. The latter statement is quite a stretch, as CF patients are colonised with a very abnormal respiratory (and GIT microbiome) and have a markedly increased burden of predominantly bacterial exacerbations.

Response: We thank the reviewer for this important point. Including clinically stable paediatric CF controls was not to claim LC-specificity per se, but to provide an additional *infection-exposed chronic respiratory disease comparator* alongside healthy children, and to test whether the key systemic signals observed in paediatric LC are robust to the composition of the comparator arm. CF controls were recruited in a stable clinical state.

To directly address the reviewer’s concern, we explicitly toned down the wording in the manuscript to avoid any implication that CF “ensures” LC-specific mechanisms, and added/expanded sensitivity analyses stratifying the control arm. Specifically, systemic cytokine levels did not differ between healthy and CF controls (new Supplementary Table 1c), supporting pooling for the cytokine/aAb assays. In addition, repeating key cytokine comparisons with LC–healthy and LC–CF contrasts yielded consistent overall conclusions

(new Supplementary Table 1d). Likewise, the main autoantibody findings were robust when analysing healthy and CF controls separately (Supplementary Fig. 4; new Supplementary Table 3.3b,c). Collectively, these analyses support that our central inferences for the reported readouts are not driven by pooling across these two control subgroups, while acknowledging that CF is not a mechanistic “baseline inflammation-free” comparator.

Supplementary Table 1c | No differences between systemic cytokine levels in Control subgroups. Comparison between Controls with vs without CF (n = 14, n = 13, respectively). Shown are two-sided p-values from Mann–Whitney U tests $\mu_{CF} \neq \mu_{No\ CF}$ (“p”) and Holm-Bonferroni-adjusted p-values (“Corrected p-value”) across all cytokines listed for this comparison. Significance level $\alpha=0.05$. Adjusted p-values were capped at 1.0.

Parameter	p	Corrected p-value	Parameter	p	Corrected p-value
IL-1 β	0.788	1	IL-5	0.522	1
GM-CSF	0.980	1	IL-13	0.471	1
IL-11	0.603	1	IL-6	0.662	1
IL-12p40	0.645	1	IL-10	0.229	1
IL-12p70	0.716	1	IFN γ	0.340	1
IL-15	0.467	1	TNF α	0.923	1
IL-18	0.458	1	IL-22	0.344	1
IL-23	0.320	1	IFN α	0.167	1
IL-27	0.054	1	IL-1 α	0.525	1

We added this clarification to the text of the main manuscript, to Materials and Methods and added new analysis in Supplementary Fig.4 and Supplementary Tables 1c, d, 3.3 b, c, to test this cohort. “The control group comprised healthy children and adolescents and paediatric cystic fibrosis (CF) patients in stable condition (Supplementary Table 1d). Control samples were available for cytokine and autoantibody assays; other readouts relied on within-cohort stratification and/or external reference datasets where applicable. Robustness to control composition was assessed by comparing healthy vs CF controls and repeating key cytokine and autoantibody analyses as LC–healthy and LC–CF contrasts; conclusions were unchanged, supporting pooling for these assays. CF served as an infection-exposed chronic respiratory disease comparator and was not intended as an inflammation-free baseline.”

6. There are a lot of assays performed and it is somewhat difficult to ascertain whether the nominally significant results reflect a priori hypotheses.

Response: We apologize for not being clear enough. The breadth of assays can make it difficult to distinguish hypothesis-driven results from nominal findings. To address this, we clarify in the revised manuscript that the assay panel was selected to interrogate a defined set of pre-specified biological axes (immune activation/inflammation, autoreactivity, and neuro-axonal injury), and we explicitly map each key readout to its corresponding pre-specified hypothesis. Specifically, we added a dedicated paragraph in Methods (Study design/Statistics) describing the pre-specified hypotheses and associated primary/secondary readouts, and labelled, at first mention in the Results, which significant readouts correspond to these pre-specified hypotheses. Analyses not tied to these pre-specified hypotheses are clearly identified as exploratory and interpreted accordingly. For transparency, we report exact p values and control for multiple testing within defined assay families (method and all adjusted p values provided in the Supplementary Tables).

7. The authors combined the cystic fibrosis and the non-cystic fibrosis controls into a single

control group on the basis of not observing markedly different IL-6 or TNF-alpha between these two subgroups. However, basing this on just two cytokines would appear to be overlooking potential differences between these two subgroups in many of the other immunological parameters investigated in this study. In addition, combining these two subgroups would seem to negate the stated aim of recruiting cystic fibrosis controls to try and distinguish long COVID-specific effects from pre-existing high infection and inflammation burden.

Response: We agree that pooling should not be justified by only two cytokines. We therefore compared healthy controls (n=14) and CF controls (n=13) across the full shared cytokine panel and basic demographics. No differences were detected (Holm–Bonferroni-adjusted $p=1.0$ for all analytes; unadjusted Mann–Whitney U tests likewise non-significant). Full statistics are provided in Supplementary Table 1c (see above).

To ensure that pooling does not obscure CF-related baseline differences relevant to our interpretation, we additionally performed sensitivity analyses stratifying the control arm. Systemic cytokine levels remained comparable between healthy and CF controls (Supplementary Table 1c), supporting pooling for the cytokine/aAb assays. Importantly, repeating the key analyses using LC–healthy and LC–CF contrasts yielded consistent effect directions and conclusions (Supplementary Table 1d). Similarly, the principal autoantibody findings were robust when analysing healthy and CF controls separately (Supplementary Fig. 4; Supplementary Table 3.3b,c).

Collectively, these analyses indicate that our central inferences for the reported readouts are not driven by pooling across the two control subgroups. We also clarify in the revised text that CF serves as a disease-control comparator rather than a mechanistic “inflammation-free” baseline.

8. Throughout, there are aspects of the analysis rationale that are unclear. The study would benefit from more consistent approach to analyses. Some specific issues:

i) Why were only three biomarkers selected for the investigating associations with lung function, when the broader range of biomarkers was used for other clinical measures? The authors highlight the relevance of IL-13 and IL-33 to lung function, but this approach seems to be quite limited.

Response: Spirometry-derived endpoints were available only in a subset of parameters and are highly correlated across parameters, which makes a broad “screen” across the full biomarker panel particularly underpowered and prone to false-positive findings once multiple testing is accounted for. We therefore limited the primary lung association analyses to a small, pre-specified set of biomarkers with strongest a priori biological and clinical relevance to airway inflammation/remodelling in paediatric respiratory disease (including IL-13 and IL-33), to keep the test family small and the inference interpretable.

To address the reviewer’s concern about consistency, we now make this rationale explicit in the Methods (Study design/Statistics) and add a complementary analysis in which lung function is tested against the broader shared biomarker panel with appropriate multiple-testing control; the full results are provided in the revised new Supplementary material (Supplementary Table 2.1, Supplementary Table 2.2 a, b), and the main text clearly distinguish the pre-specified (primary) lung hypotheses from the broader sensitivity screen.

ii) Analyses are presented for testing differences between the control group and the LC <1 year group, and investigating differences between the LC <1 yr and LC 1-3 yr group, but not between the control and LC 1-3 yr groups. This seems like a notable omission.

Response: We agree that not reporting the direct control vs LC 1–3 years comparison reduced information. In the revision, we therefore add this missing pairwise contrast for the shared serum readouts (cytokines and autoantibodies) using the same covariate-adjusted workflow as for the other between-group comparisons (Fig.1-3, Supplementary Fig.1,2,4). Results are now reported alongside the existing contrasts in the Results and corresponding Supplementary Tables; conclusions remain unchanged.

Fig for referee’s use: Example of visualisation displaying all groups combined in Fig.1-3, here Fig. 1d. Serum IFN α in controls and LC stratified by TsinceIndex (< 1 yr; 1-3 yr). Boxplots show median and IQR. Kruskal–Wallis with Dunn’s post-hoc tests; within-LC time-window comparison by linear mixed-effects modelling (LMM)(participant ID as random intercept).

iii) There are inconsistencies in covariates included in models (notably, sex is adjusted for in some sets of models but not others), it would be helpful for authors to more clearly articulate how they selected their model and the differences. Another inconsistency is how data from participants with multiple visits were handled, with some analyses using the mean of these visits (e.g., Figure 2), others using a ‘single representative value’ (e.g., Figure 1) and others treating each observation as independent despite this approach biasing the presented analysis (e.g., the Spearman correlations in Supplementary Table 2.1c).

Response: We have revised the statistical analysis plan to make model selection, covariate adjustment, and the handling of repeated visits explicit and consistent across the manuscript. Ruels are decribed in Methods.

For cross-sectional comparisons that evaluate a given time window against controls only, we now use one value per participant per window; if a participant contributed two visits within the same window, these were collapsed to the participant-level mean to avoid pseudo-replication and inflated precision.

All within-LC analyses, comparing early versus later time windows explicitly leverage the repeated-measures structure and are performed using linear mixed-effects models (LMMs) with participant ID as a random intercept. In this line, we revised the correlation analysis underlying the former Supplementary Table 2.1c (now Supplementary Table 3.1) to follow the same participant-level rule set, and the Methods/figure legends now clearly distinguish window-based cross-sectional evaluations from LMM longitudinal analyses.

According to Fig. 1a-c: Participant demographics were assessed at **study entry/enrolment** and are time-invariant within the study, and symptom-based measures shown in Fig. 1 were likewise collected at enrolment (i.e., reflecting the clinical status at study entry rather than longitudinal change). Therefore, restricting Fig. 1a-c to Visit 1/enrolment provides the appropriate and internally consistent representation of baseline characteristics; longitudinal changes are analysed separately in the LMM frameworks described in the Methods and shown in the corresponding figures Fig. 1e and Supplementary Fig. 1. For clarification, wording is changed to enrolment/study entry and follow-up.

In response to the reviewer's comment on covariate consistency, we implemented a harmonised adjustment strategy and conducted sensitivity analyses including a set of clinically plausible covariates (comorbidity status, infections prior long COVID, vaccination, time since index infection (TsinceIndex), age, and sex). Inclusion of these covariates did not materially change effect estimates or conclusions for LC severity, and in extended mixed-effects models for Fig. 2d and Fig. 4 the added covariates did not improve model fit and did not alter the primary associations; full outputs are provided in the revised Supplementary material (including new Supplementary Table 1b). Accordingly, we present the parsimonious primary models in the main figures and text, and provide the fully adjusted models as sensitivity analyses.

Together, these changes make model selection explicit, align covariate adjustment across analyses, and ensure that no inference relies on treating repeated measurements as independent observations.

9. The value of the k-means approach used to define clusters of FEV1 z-scores is not well-articulated, with the decision to define three clusters appearing largely arbitrary. Could the authors please provide additional information on this approach including the metrics they used to validate their approach?

Response: We agree that the k-means clustering of FEV1 Z-scores (including the choice of three clusters) was not sufficiently informative to justify inclusion as a main analytic element. In the revision, we therefore remove the k-means clustering analysis and instead present our finding as lung function using a transparent continuous-data framework. Specifically, we now show the distribution of FEV1 z-scores across predefined TsinceIndex windows (<1 year vs 1–3 years), and we indicate the lower limit of normal ($z \leq -1.64$) directly in the figure (Fig. 2b). To formally account for repeated visits and potential clinical/exposure covariates, we analyse FEV1 z-scores using a linear mixed-effects model (LMM) with participant ID as a random intercept and fixed effects including comorbidity, sex, age, VOC wave, infections, vaccinations, and disease duration/time window (Supplementary Table 2.1).

This revised presentation provides a clearer and more reproducible picture of pulmonary function in the cohort: FEV1 z-scores are within the normal range, with only a minority below the lower limit of normal, while retaining substantial inter-individual variability (Results; Fig. 2b).

New Fig. 2b. Boxplot of forced expiratory volume in 1 s (FEV1) z-scores across time windows (participant-level summaries per window; see Methods). The grey band denotes the normal reference range (± 1.64). Group differences were tested using linear mixed-effects models (LMMs) participant ID as a random intercept.

10. The lung function analyses do not appear to be discussed at all in the discussion section. The authors' discussion of these analyses would be beneficial.

Response: As it was originally written for the Nature format, the discussion was too brief. We have now added an explicit discussion of the lung function findings. In the revised Discussion, we now summarise that spirometry was largely within the normal range at the cohort level (with

only a minority below the lower limit of normal), consistent with the interpretation that routine spirometry does not explain the high burden of dyspnoea and fatigue in this paediatric LC cohort. We further discuss the observed association between higher FEV1 z-scores and systemic IL-13, highlighting that repair-oriented programmes may operate alongside pro-inflammatory mechanisms and may contribute to inter-individual heterogeneity even when values remain within population norms.

11. The discussion would also benefit from clearer discussion of this study's limitations and strengths, which seem to have been omitted.

Response: Limitations and strength has been added to the discussion, now. One paragraph and within other text blocks in the discussion.

Some other more minor comments:

12. Introduction: "Autoimmune diseases were found to occur with greater frequency" – is this referring to autoimmune diseases as a risk factor of LC, or autoimmune diseases as a consequence of LC?

Response: We revised that line to "refer explicitly to autoimmune disease as a post-infectious outcome (de novo autoimmune diagnoses after SARS-CoV-2 / in post-COVID cohorts), not as a pre-existing risk factor for developing LC."

In the current Introduction, the wording now reads along the lines of: "Autoimmune phenomena have been reported more frequently in paediatric post-COVID cohorts... 1.1% developed de novo autoimmune diseases ..." and then we contextualise that this may motivate looking for autoreactivity-related signatures in subsets, without implying that LC uniformly reflects overt systemic autoimmunity.

13. Results: "This indicates that different pathological mechanisms contribute to disease severity and progression." – is this overstating the causal evidence of these associations?

Response: We exchanged the sentence with:and patients followed heterogeneous trajectories (improving vs worsening), consistent with concurrent protective and pathogenic immune arms whose relative dominance shifts longitudinally.

14. Results: why was IL-13 selected over IL-33 for the lung function models?

Response: IL-13 was used because it was the pre-specified lead marker for the type-2/repair axis in our serum cytokine panel and showed the clearest, most robust association pattern with spirometry in our data. IL-33, in contrast, is an upstream epithelial "alarmin" that is often low/variable in peripheral blood and in our dataset did not provide additional explanatory value beyond the downstream type-2 effector signal captured by IL-13. We now clarify this rationale in the Results/Methods and, for transparency, report the corresponding IL-33 spirometry analyses in the Supplementary tables.

15. Table 1 + 2: What was the rationale for including interaction terms in the models presented in Table 2 but not the models presented in Table 1?

Response: Interaction terms were included in Table 2 (now Table 3) because these models were explicitly specified to test effect modification, i.e., whether associations differ by longitudinal trajectory and/or time window; this is directly captured by predictor \times trajectory (or predictor \times time) interaction terms. In contrast, the models in Table 1 were specified to estimate average between-group differences (main effects) under a parsimonious adjustment strategy. Adding interactions to these cross-sectional comparisons would increase model complexity and reduce interpretability and power in the absence of a pre-specified hypothesis for effect modification. We now state this model-selection rule explicitly in the Methods.

16. Methods: “The z-score represents the number of standard deviations from the mean of healthy age-matched controls” – this is largely duplicative text with the previous sentence.

Response: Thank you for carefully reading our manuscript. We have omitted the sentence accordingly.

17. Supplementary Figure 2 + 3.1: it is not clear what the different panels refer to.

Response: We clarified this issue. Thank you for pointing it out. (We also attached a list of all tables and Figures.)

18. EDT6 a + b – Why has information on fixed effects been provided only for the top models?

Response: We have revised Supplementary Table 6a,b to improve clarity. We now explicitly indicate which candidate models were tested but did not show evidence of an association for the primary outcome. For these non-supported models, we do not report the full set of fixed-effect coefficients in the table to avoid disproportionate emphasis on numerous uninterpretable estimates and to limit multiple comparisons in narrative interpretation. Full fixed-effect outputs for all candidate models are available upon request. We therefore present detailed fixed-effect estimates only for models with evidence of an association, where coefficients are interpretable in the context of the study question.

Reviewer #2 (Remarks to the Author):

Reviewer #3 (Remarks to the Author):

Response: We thank the reviewer for the constructive assessment and for highlighting the importance of paediatric LC research. We have revised the manuscript to present a clearer, hypothesis-driven framework, to consolidate the timing/control backbone across Figures 1–3, and to clarify which analyses are primary versus exploratory. We respond to each comment in detail below.

Reviewer #3 General comments: In this article titled “Immune-Metabolic Programs Drive Disease Trajectories in Paediatric Long COVID”, the authors describe aspects of both the clinical course and underlying biological differences in pediatric patients diagnosed with long COVID. Using a combination of survey data, blood analyte and cytokine profiling, as well as statistical analysis the authors describe finding differences both when comparing pediatric Long COVID patients to controls in addition to three different biologically distinct sub-categories of Long COVID. These findings are novel to the field. Pediatric Long COVID is an important area of study as there are significantly fewer studies evaluating the pediatric population compared to adults. As the authors point out, there appear to be different mechanisms involved, thus underscoring the need to study pediatric patients directly. Overall, the study design, the methods, and analysis are generally well constructed. My most significant criticism is that the rationale for many of the experiments done should be more clearly explained. There are many different hypotheses including Th2 vs Th17 immune skewing, EBV serostatus, neuro-axonal injury, and coagulopathies all with the added complexity of timing of Long COVID illness. Clearly these are all important aspects to evaluate, but the study would be stronger if this was laid out in a more logical fashion for the reader to follow.

Response: We thank the reviewer for the constructive assessment and agree that the experimental rationale and its timing component should be easier to follow. We therefore made our a priori framework explicit in the Introduction (temporally dynamic immunometabolic endotypes; p. 4, l. 79–89). In addition, we added a study design/phenotyping overview (Fig. 1a) and we now present the control comparison together with the two pre-defined disease windows (<1 yr; 1–3 yr since index infection) as a consistent backbone across Fig. 1d, Fig. 2 and Fig. 3 (Fig. 1 legend p. 26, l. 657–666; Fig. 2 legend p. 28, l. 670–688; Fig. 3 legend p. 29, l. 694–708). We also clarified how TsinceIndex windows are defined and how repeated visits are handled for cross-sectional windowed contrasts versus within-LC longitudinal models (Methods p. 37, l. 892–905).

(ii) the severity model in which covariates were defined a priori to represent complementary biological axes (p. 9, l. 197–210), and (iii) NfL and autoantibodies framed as pre-specified exploratory axes to probe heterogeneity (p. 9, l. 217–224). Together, these changes present the experimental logic in a more linear, hypothesis-driven manner and clarify how timing is integrated throughout the manuscript.

Figure 1/Demographics:

1. Including criteria of how Long COVID was diagnosed and how participants were selected: The methodology doesn't describe how the Long COVID patients were selected/diagnosed. As a clinical diagnosis, it would be helpful to understand how the clinicians determined if participants have Long COVID. Did they follow the WHO criteria? Where were they diagnosed and followed? Did families/children volunteer themselves?

Response: We thank the reviewer and agree that clinical ascertainment and participant selection should be described explicitly, what we did now. Children and adolescents with suspected post-COVID-19 condition were recruited through the dedicated Long-COVID outpatient clinic at Jena University Hospital and were referred by board-certified paediatricians (participants did not self-enrol from the general community). At enrolment, clinicians obtained a structured history that explicitly captured pre-existing symptoms and comorbidities prior to SARS-CoV-2 infection; eligibility required that symptoms used for the case definition first appeared after a confirmed infection, interfered with daily activities, and informed consent was

obtained from both the child/adolescent and a parent/guardian. Case definition followed the NICE rapid guideline on long-term effects of COVID-19 (Dec 2020; updated Nov 2021). Of 106 patients evaluated in clinic, 78 met prespecified criteria and were proposed for inclusion; all proposed cases were re-reviewed by the study lead for diagnostic consistency, and one candidate was excluded because relevant symptoms clearly pre-dated infection. The final analysis set comprised 74 participants with complete data at both visits. We have clarified these procedures in the Methods (p. 31–32, l. 738–756).

Participants were assessed at the clinic at study enrolment Visit 1 and re-assessed 3–6 months later (follow-up). Of note, during much of the recruitment period, school-aged children underwent routine school-based SARS-CoV-2 screening with PCR confirmation of positive rapid antigen tests, supporting high-confidence documentation of index infection dates.

2. Number of participants needed: How was 73 participants selected? Was a power calculation done? I see on the clinical trials website, the authors were looking to recruit 150 participants. How were the number of controls selected?

Response: We thank the reviewer for this question. No formal a priori power calculation was performed for this multi-axis immune–metabolic profiling study, because paediatric LC effect sizes and variance components were insufficiently established at study initiation and because the design spans multiple correlated outcomes and time windows, with analyses organised around pre-specified primary objectives complemented by exploratory modules (Methods p. 37, l. 885–890). Accordingly, recruitment was feasibility-based within the clinic programme registered in DRKS (DRKS00028523; Methods p. 31, l. 735–736). The cohort size reported here reflects consecutive eligible referrals who, within the recruitment period and clinical/study capacity, completed the two-visit protocol with full deep phenotyping: 106 patients were evaluated, 78 met prespecified criteria and were proposed for inclusion, one was excluded after adjudication due to symptoms clearly pre-dating infection, and 74 completed full data collection at both visits and were included in the final analyses (Methods p. 31, l. 752–756). Controls were recruited independently and comprised healthy children/adolescents and clinically stable paediatric cystic fibrosis patients ($n = 27$; Methods p. 32, l. 758–768; Supplementary Table 1a,d). Control biospecimens of control group children were available for cytokine and autoantibody assays; for several other readouts we relied on within-cohort stratification and/or external reference datasets where appropriate. Robustness to control composition was assessed by repeating key cytokine/autoantibody contrasts using healthy and CF controls separately; conclusions were unchanged, supporting pooling for these assays (Methods p. 32, l. 762–768; Supplementary Table 1c,d).

3. Age difference in control vs LC: The control group is younger and not age matched—which is something that might matter when discussing EBV related exposures.

Response: We agree that age is a potential confounder for EBV-related analyses given the strong age dependence of EBV seroprevalence. Controls were slightly younger than LC participants (mean \pm SD 10.9 \pm 5.15 vs 14.1 \pm 2.5 years; Mann–Whitney U, $p=0.07$; Supplementary Table 1a). Importantly, our EBV findings are derived from within-cohort analyses in the LC group, modelling EBV exposure (anti-EBV EBNA) as a covariate in LMMs while adjusting for age (as well as sex, TsinceIndex, vaccination status and comorbidity), rather than from EBV case–control comparisons (Methods p. 32, l. 758–766; Results p. 11, l. 268–276; Methods p. 37, l. 906–912).

4. Control group COVID-19 status: I am concerned that 61% of the control patients analyzed had not had SARS-CoV-2. There is mounting evidence that any SARS-CoV-2 infection alters the immune system regardless of LC status. Do the authors findings still stand when the Long COVID group is compared to convalescent controls? This would be important to determine.

Response: We agree that the most informative comparator would be a paediatric convalescent (post-SARS-CoV-2, non-LC) control group, because SARS-CoV-2 infection alone may induce persistent immune alterations. In our control cohort, prior COVID-19 history was captured by parent report (yes/no/not reported: 10/10/7; Supplementary Table 1a), so a strictly convalescent-only comparator is limited in size (n = 10) and incomplete for a substantial fraction of controls.

Given these constraints, we did not base our primary conclusions on “SARS-CoV-2-naïve vs convalescent” control stratification. Instead, we assessed robustness to control composition using the two available control subgroups (healthy vs clinically stable CF), repeating key cytokine and autoantibody comparisons as LC-healthy and LC-CF contrasts; conclusions were unchanged, supporting pooling for these assays (Methods p. 32, l. 758–766; Supplementary Table 1d; Supplementary Fig. 4).

We have clarified this limitation and the corresponding robustness analyses in the Methods (Methods p. 32, l. 762–766; p. 37, l. 913–917).

5. Chronic Disease prior to LC status: There are a number of LC patients who have other chronic illnesses. Should they be included in this analysis (ie: these diseases do not confound the data)? If they should be included, the rationale should be explained.

Response: We agree that pre-existing chronic disease could confound associations. Because the cohort was designed to represent children and adolescents presenting with paediatric LC in routine care, we did not exclude participants with stable, pre-existing major chronic disease (new Supplementary Table Pre-existing conditions in paediatric long COVID: cohort prevalence versus reported population benchmarks). To address potential confounding, all primary models adjusted a priori for comorbidity status, defined as a binary indicator (yes/no) capturing clinically significant pre-existing diagnoses with potential systemic relevance. We did not include common, often mild and heterogeneously ascertained conditions (e.g., questionnaire-based self-reported allergic history, food intolerances/malabsorption, or neurodevelopmental diagnoses such as ADHD/learning disorders) in this covariate to avoid over-adjustment and misclassification. Adjustment for major comorbidity did not change effect directions and did not alter statistical significance for the primary outcomes, indicating that the main findings are unlikely to be driven by pre-existing chronic disease (see Supplementary Table 7 for a detailed breakdown).

Supplementary Table 7 | **Pre-existing conditions in paediatric LC: cohort prevalence versus reported population benchmarks.** Pre-existing conditions recorded at enrolment among participants with paediatric LC (n = 74). Data are shown as n (%) unless stated otherwise. Published prevalence estimates are provided for contextual comparison only and should be interpreted with caution given heterogeneity in age strata, case definitions, data sources (population-based surveys vs clinical cohorts), and ascertainment; they were not used for statistical inference in this study. Benchmark estimates were obtained from population-

based German health surveys, international systematic reviews, and disease-specific registries (e.g., Robert Koch Institute; KiGGS Wave 2, 2014–2017, published 2018). Participants could report more than one pre-existing condition; therefore, counts across conditions are not mutually exclusive and may sum to > 74. “Atopy” reflects questionnaire-based self-report and was not necessarily physician-diagnosed.

Group	Condition	Cohort n (%)	Population benchmark (context only)
Neurodevelopmental / Neurological	ADHD (Attention Deficit Hyperactivity Disorder)	4 (5.4%)	~4.4 % (diagnosed cases, DE)
	Dyslexia	2 (2.7%)	1.9–2.6 % strict, up to 7–15 % broadly defined
	Migraine	4 (5.4%)	~10 % (children/adolescents)
	Other (each n=1; 1.4%)	2 (2.7%)	variable / referral-based (see note)
Gastrointestinal	Lactose Malabsorption	7 (9.5%)	~20–40% in clinical groups, lower overall
Metabolic / Endocrine	Obesity	2 (2.7%)	5.9 % (obesity), 15.4 % including overweight
	Other (each n=1; 1.4%)	2 (2.7%)	study-dependent (see note)
Cardiac	Other (each n=1; 1.4%)	2 (2.7%)	study-dependent (see note)
Respiratory–Allergic	Bronchial Asthma	3 (4.1%)	3.5–4 % (12-month prevalence)
	Self-reported atopy (questionnaire-based)	27 (36%)	≥16 % with ≥1 atopic disease
Other / rare conditions	Other (each n=1; 1.4%)	9 (12.2%)	variable / often no robust prevalence (see note)
	Overall	Cystic Fibrosis	0 (0%)
	No Prior Illness	37 (50.0%)	n/a

Notes (what is included in “Other (each n=1; 1.4%)”)

- **Neurodevelopmental / Neurological (2x n=1):** Sensory processing disorder; psychogenic gait disorder.
- **Metabolic / Endocrine (2x n=1):** NAFLD; Hashimoto’s thyroiditis.
- **Cardiac (3x n=1):** Coronary fistula; bicuspid aortic valve; mild mitral regurgitation.
- **Other / rare (6x n=1):** Gilbert’s syndrome (Gilbert–Meulengracht syndrome); FASD; Loeys–Dietz syndrome; scoliosis; acne; visual disorders (amblyopia); divergent strabismus; myofascial pain syndrome; chromosome 13 anomaly (Trisomy 13).

Clinical Features of Long COVID

1. FEV1 z-score: It is interesting that the K modes analysis of FEV1 z-score showed 3 different groups. Was there any additional analysis done on these 3 groups (ie; in relation to blood analytes/cytokines)? If so, how did these compare to the control group?

Response: We previously explored an unsupervised k-modes clustering of FEV1 z-scores, but we agree that “clusters” defined by a single pulmonary variable are not readily interpretable as biological endotypes and do not add beyond presenting the continuous distribution. We therefore removed the k-modes grouping and now report FEV1 as age/sex/height-standardised z-scores across the pre-defined time windows (Fig. 2b) and focus on hypothesis-driven modelling of inter-individual heterogeneity. To link lung function to systemic biology, we analysed FEV1 z-scores as a continuous outcome in linear mixed-effects models (random intercept: patient ID) using a pre-specified airway/type-2 versus inflammatory framework. This identifies IL-13 (positive) and IL-6 (negative) as independent correlates of FEV1 z-scores, robust to adjustment for pre-specified covariates (Fig. 2d; Supplementary Table 2.2). Regarding controls, spirometry was not performed as a control readout; instead, FEV1 is expressed relative to established paediatric reference norms (z-scores), which provides the comparator framework for lung function itself. Systemic cytokine levels, however, are compared directly to controls (Fig. 2c) and then related to FEV1 within the LC cohort (Fig. 2d).

New Fig. 2b. Boxplot of forced expiratory volume in 1 s (FEV1) z-scores across time windows (participant-level summaries per window; see Methods). The grey band denotes the normal reference range (± 1.64). Group differences were tested using linear mixed-effects models (LMMs) participant ID as a random intercept.

Figures 2 and 3

1. Is there a reason that the LC>1 year and control plots are separated out from LC>1 year? Could they be all on the same plots? If there is a reason the data is separated out this way, it would benefit from further explanation.

Response: We agree that displaying all groups on the same plots improves clarity and supports direct comparison across time windows and controls. In the revised manuscript, we therefore restructured Figs. 2 and 3 so that controls and both pre-defined LC windows (<1 yr; 1–3 yr since index infection) are shown together within each panel wherever the readout is shared, and we added the previously missing pairwise contrasts using the same covariate-adjusted workflow as for the other between-group comparisons. The corresponding results are now reported in the Results and in the updated Supplementary Tables/Figures; conclusions remain unchanged (Fig. 1d, 2–3; Supplementary Table 2.*; Supplementary Fig. 4).

Fig for referee’s use: Example of visualisation displaying all groups combined in Fig.1-3, here Fig. 1d. Serum IFN α in controls and LC stratified by TsincelIndex (< 1 yr; 1-3 yr). Boxplots show median and IQR. Kruskal–Wallis with Dunn’s post-hoc tests; within-LC time-window comparison by linear mixed-effects modelling (LMM)(participant ID as random intercept).

2. The fact that statistical significance is represented after multiple corrections analysis should be more clearly stated in the figure legends. If those are the effects that have survived after correcting for 43 analytes, that is much more compelling.

Response: Yes, indeed, all significance indicators shown reflect results that remained significant after multiple-testing correction across the 43 analytes. We have now stated this explicitly in the relevant figure legends and in the table headings/text to ensure it is clear that the highlighted associations represent findings that survived correction for 43 comparisons.

Figure 4

This is a nice visual representation of a lot of data. It would be nice if there was a way to have some of the data from Table 1 as a separate panel in this figure also. Maybe even just a plot of EBV EBNA IgG level in the Long COVID group vs control.

Response: Thank you for appreciating this range of data. We apologise for not being clear enough about the visual representation of the Table 2 (formerly Table 1) results. The EBV-related results are indeed visualised in Fig. 4 as the set of covariate-adjusted LMM associations with EBV exposure, with full model outputs provided in Table 2 and Supplementary Tables 4–5. We have now made this link explicit in the Fig. 4 legend and Table 2 caption and clarify that only associations surviving Holm–Bonferroni correction are displayed (Fig. 4; Table 2 caption; Methods p. 37, l. 885–890). In addition, anti-EBV EBNA IgG titres and EBV-naïve/experienced stratification are shown directly in Fig. 2, providing the requested visualisation of EBV serology alongside other core stratifiers (Fig. 2; Fig. 2 legend p. 28, l. 670–688).

Discussion:

In general, the discussion section would benefit from expanded rationale and thoughts regarding the different endotypes the authors describe. The comparison to adult LC is important however the manuscript would benefit from more explanation about their own findings. Are there age/sex contributions to each of those 3 types found? How does the development of the pediatric immune system progress over time—what is already known? What is known in relation to other chronic infections? How does EBV status shape immune system? There are some known age-related immune changes such as skewing from Th1 to Th2. How does this play into the authors' hypotheses and findings?

Response: We agree and have expanded the Discussion to more explicitly interpret the three endotype axes as partially independent programmes that can co-occur and shift over time, linking cross-sectional contrasts to longitudinal trajectories and clarifying how this differs from a simple linear inflammation–severity model (p. 14–15, l. 345–370). To address potential demographic contributions, we tested age and sex effects on the endotype axes and key cytokine shifts using covariate-adjusted mixed models; we observed no consistent stratification by age or sex, and adjustment did not materially change endotype–symptom associations (p. 16, l. 393–406).

We also added a concise, age-informed immunology context, including developmental tuning across childhood/adolescence and puberty-associated modulation, and discuss how these principles align with our observation of a persistent Th2-leaning/innate programme alongside a waning early antiviral axis (Introduction p. 3–4, l. 62–71; Discussion p. 14–16, l. 361–392). Finally, we strengthened the rationale around chronic/latent infections by clarifying that EBV reactivation appeared rare in this cohort (and children in general), and we interpret EBV serostatus as a stable imprint of prior herpesvirus experience that can shape immune set-points and stratify an innate-cytokine/neutrophil-biased programme, rather than implying EBV reactivation as a dominant driver (p. 16–17, l. 406–415).

Response to Reviewers

We thank the Reviewers for their careful evaluation of our manuscript and for their constructive comments. We have therefore revised the manuscript accordingly and provide a detailed point-by-point response below.

Reviewer #1

Comment: Thank you for the opportunity to review the revised submission. The authors have undertaken considerable revision of this manuscript to address the previous reviews, and in doing so have clarified some of the concerns and details around the methodology of this study.

Response: We thank the reviewer for this thoughtful and constructive reassessment of our revised manuscript and for acknowledging the substantial revisions undertaken. We also appreciate the remaining comments regarding the overall clarity and messaging of the manuscript, which we address below.

Comment: The sample size is modest and some of the large number of analyses are difficult to interpret with confidence.

Response: We thank the reviewer for this important comment. We agree that the modest sample size and the broad range of analyses require cautious interpretation. However, we would like to stress that this study is, to our knowledge, among the most comprehensively and deeply phenotyped prospective paediatric long COVID cohorts reported to date. Key strengths include the prospective longitudinal design, the long duration of illness in the affected children, and the follow-up assessment, given that comparable studies are often limited by shorter observation periods and less detailed phenotyping. Importantly, the use of mixed linear effects

models (LMM) constitutes a methodological innovation and strength of the study, as it allows the longitudinal repeated-measures structure of the data to be modelled appropriately while accounting for within-subject variability over time. Although the number of participants is modest, the longitudinal design yields a substantially larger number of observations ($n = 148$), which increases the effective information content for model estimation. We therefore believe that our approach provides robust and novel longitudinal insights that are currently scarce in paediatric long COVID research.

Comment: The incorporation of additional analyses has better supported the approach taken to control groups in this study, although the use of CF patients introduces complexities in interpretation of the findings.

Response: We thank the reviewer for this important comment. We agree that the inclusion of children with cystic fibrosis (CF) as a control group adds complexity to the interpretation. To address this, we compared key findings with the non-CF control subgroup separately and obtained comparable results, as shown by sensitivity analyses using stratified control subgroups (Supplementary Table 1c,d; Supplementary Table 3.3a–c; and Supplementary Fig. 4). Direct comparisons between the healthy and CF control subgroups likewise did not reveal relevant differences in the assessed inflammatory measures (Supplementary Table 1c). CF controls were included as a clinically relevant comparator group with chronic disease and infection-related burden, complementing the healthy control group. At the same time, the control groups were used primarily for contextualization. The main conclusions of the study are based on the longitudinal analyses within the LC patient cohort, particularly the linear mixed-effects models (LMMs), which assess within-subject changes over time and support the central interpretation of the findings independently of the specific control composition.

Comment: The authors have also indicated throughout where analyses were based on primary hypotheses or were exploratory/hypothesis-generating. However, even with these revisions, the messaging of the manuscript is made less clear by the broad range of hypothesis-driven and hypothesis-generating analyses across different sets of data and comparisons to investigate the various aspects of the authors' aims.

Response: We thank the reviewer for this important comment. We agree that the conceptual organisation of the analyses required clearer articulation. Our study was not designed as a collection of independent exploratory analyses, but rather around a pre-specified framework of biologically motivated axes of disease heterogeneity. Specifically, we structured the analyses to capture three complementary dimensions of paediatric long COVID biology: (i) a temporal axis reflecting disease phase (TsincelIndex), (ii) an immune imprinting axis related to prior EBV exposure, and (iii) an autoreactivity-associated axis defined by anti-DFS70 status. Within each of these axes, we evaluated pre-defined sets of immune, metabolic and clinical readouts, rather than testing unrelated hypotheses. Importantly, these axes were not assumed to follow a strict hierarchy but were considered partially independent and potentially interacting dimensions of disease biology, consistent with the heterogeneity observed in paediatric LC. We have revised the manuscript to make this conceptual structure more explicit, particularly by clarifying how individual analyses relate to these overarching axes and by introducing this framework earlier in the Introduction, Results and Discussion.

Comment: There are multiple immune and endotype 'axes' considered in this manuscript, and while the authors highlight specific associations of interest in the discussion from the milieu of analyses undertaken, the importance of these findings for the field or in the context of potential opportunities for prognosis or management are not sufficiently articulated.

Response: We thank the reviewer for this important comment. We agree that the importance of the identified immune and clinical axes for the field, and their possible implications for

prognosis and clinical management, were not sufficiently articulated in the original Discussion. We have therefore revised the Discussion to clarify that the principal contribution of our study is to provide a biologically structured, longitudinal framework for paediatric long COVID rather than a single cross-sectional severity model. Specifically, we now emphasize that these axes reflect partially independent, temporally evolving programmes rather than fixed patient “types”, and may help distinguish pathways linked to persistent symptom burden from those associated with recovery or resilience. We further state that, if validated in independent cohorts, this framework could support risk stratification, longitudinal monitoring, and the design of subgroup-informed observational and interventional studies. In addition, we now highlight that several candidate markers emerging from these axes are routinely accessible in clinical practice, which may facilitate future translation into pragmatic prognostic and monitoring strategies after external validation. We also revised the Discussion to more clearly address implications for clinical evaluation. In particular, we now note that negative findings can also help refine diagnostic pathways; for example, the absence of consistent cardiac involvement in our cohort suggests that extensive investigations such as routine echocardiography may not be required in all paediatric long-COVID evaluations once major pathology has been excluded. To improve the conceptual clarity of this framework throughout the manuscript, we also revised the end of the Introduction and the Results to state explicitly that the analyses were structured along predefined, biologically motivated dimensions of disease phase (TsinceIndex), EBV exposure, and autoreactivity (anti-DFS70), while integrating complementary clinical, immunological, metabolic, and cardiovascular readouts.

For example, the Results section now states: “To interrogate paediatric LC heterogeneity, we structured the analyses along predefined, biologically motivated dimensions of disease phase (TsinceIndex), EBV exposure, and autoreactivity (anti-DFS70), and assessed complementary clinical, immunological, metabolic, and cardiovascular readouts within this framework.”

Reviewer #2 (Remarks to the Author):

Comment: I co-reviewed this manuscript with one of the reviewers who provided the listed reports. This is part of the Nature Communications initiative to facilitate training in peer review and to provide appropriate recognition for Early Career Researchers who co-review manuscripts.

Response: We thank the co-reviewer for their contribution to the evaluation of our manuscript.

Reviewer #3 (Remarks to the Author):

Comment: This submission is significantly improved after the modifications the authors have made. Their rationale for study design and subsequent analysis is clear to follow. The figures are also much improved and each have a clear message. My only small edit is that in table 1, I believe the second half with corrected p values should have one part that is C vs LC <1yr and C vs LC >1 yr (it currently has both as >1yr). I thoroughly enjoyed reading the discussion and appreciate the idea of the three pediatric LC endotypes identified are mechanistic based and dynamic. This is an important concept for the field to consider. I have no additional questions or concerns.

Response: We thank the reviewer for this very positive and thoughtful assessment of our revised manuscript. We are grateful for the encouraging comments regarding the study design, analysis, figures, and discussion, and we appreciate the recognition of the concept of dynamic, mechanistically informed paediatric long COVID endotypes.

We thank the reviewer for noting the error in Table 1. This has now been corrected.